# Learning Trajectories are Generalization Indicators

**Jingwen Fu**[1]*, **Zhizheng Zhang**[2]†, **Dacheng Yin**[3]*, **Yan Lu**[2] , **Nanning Zheng**[1]†
fu1371252069@stu.xjtu.edu.cn
{zhizzhang,yanlu}@microsoft.com
ydc@mail.ustc.edu.cn
nnzheng@mail.xjtu.edu.cn
[1]National Key Laboratory of Human-Machine Hybrid Augmented Intelligence,
National Engineering Research Center for Visual Information and Applications,
and Institute of Artificial Intelligence and Robotics, Xi'an Jiaotong University,
[2]Microsoft Research Asia, [3]University of Science and Technology of China

## Abstract

This paper explores the connection between learning trajectories of Deep Neural Networks (DNNs) and their generalization capabilities when optimized using (stochastic) gradient descent algorithms. Instead of concentrating solely on the generalization error of the DNN post-training, we present a novel perspective for analyzing generalization error by investigating the contribution of each update step to the change in generalization error. This perspective enable a more direct comprehension of how the learning trajectory influences generalization error. Building upon this analysis, we propose a new generalization bound that incorporates more extensive trajectory information. Our proposed generalization bound depends on the complexity of learning trajectory and the ratio between the bias and diversity of training set. Experimental observations reveal that our method effectively captures the generalization error throughout the training process. Furthermore, our approach can also track changes in generalization error when adjustments are made to learning rates and label noise levels. These results demonstrate that learning trajectory information is a valuable indicator of a model's generalization capabilities.

## 1  Introduction

The generalizability of a Deep Neural Network (DNN) is a crucial research topic in the field of machine learning. Deep neural networks are commonly trained with a limited number of training samples while being tested on unseen samples. Depite the commonly used independent and identically distributed (i.i.d.) assumption between the training and testing sets, there often exists a varying degree of discrepancy between them in real-world applications. Generalization theories study the generalization of DNNs by modeling the gap between the empirical risk [37] and the popular risk [37]. Classical uniform convergence based methods [20] adopt the complexity of the function space to analyze this generalization error. These theories discover that more complex function space results in a larger generalization error [38]. However, they are not well applicable for DNNs [33, 22]. In deep learning, the double descent phenomenon [6] exists, which tells that larger complexity of function space may lead to smaller generalization error. This violates the aforementioned property in uniform convergence methods and imposes demands in studying the generalization of DNNs.

Although the function space of DNNs is vast, not all functions within that space can be discovered by learning algorithms. Therefore, some representative works bound the generalization of DNNs based

---

*Work done during internships at Microsoft Research Asia.
†Corresponding Authors

on the properties of the learning algorithm, *e.g.*, stability of algorithm [11], information-theoretic analysis [40]. These works rely on the relation between the input (*i.e.*, training data) and output (weights of the model after training) of the learning algorithm to infer the generalization ability of the learned model. Here, the relation refers to how the change of one sample in the training data impacts the final weights of model in the stability of algorithms while referring to the mutual information between the weights and the training data in the information-theoretic analysis. Although some works [24, 11] leverage some information from training process to understand the properties of learning algorithm, there is limited trajectory information conveyed.

The purpose of this article is to enhance our theoretical comprehension of the relation between learning trajectory and generalization. While some recent experiments [9, 13, 29] have shown a strong correlation between the information contained in learning trajectory and generalization, the theoretical understanding behind this is still underexplored. By investigating the contribution of each update step to the change in generalization error, we give a new generalization bound with rich trajectory related information. Our work can serve as a starting point to understand those experimental discoveries.

## 1.1 Our Contribution

Our contributions can be summarized below:

- We demonstrate that learning trajectory information serves as a valuable indicator of generalization abilities. With this motivation, we present a novel perspective for analyzing generalization error by investigating the contribution of each update step to the change in generalization error.

- Utilizing the aforementioned modeling technique, we introduce a novel generalization bound for deep neural networks (DNNs). Our proposed bound provides a greater depth of trajectory-related insights than existing methods.

- Our method effectively captures the generalization error throughout the training process. And the assumption corresponding to this method is also confirmed by experiments. Furthermore, our approach can also track changes in generalization error when adjustments are made to learning rates and label noise levels.

## 2 Related Work

**Generalization Theories**   Existing works on studying the generalization of DNNs can be divided into three categories: the methods based on the complexity of function space, the methods based on learning algorithms, and the methods based on PAC Bayes. The first category considers the generalization of DNNs from the perspective of the complexity of the function space. Many methods for measuring the complexity of the function space have been proposed, *e.g.*, VC dimension [39], Rademacher Complexity [4] and covering number [33]. These works fail in being applied to DNN models since the complexity of the function space of a DNN model is too large to deliver a trivial result [41]. This thus motivates recent works to rethink the generalization of DNNs based on the accessible information in different learning algorithms such as stability of algorithm [11], information-theoretic analysis [40]. Among them, the stability of algorithm [7] measures how one sample change of training data impacts the model weights finally learned, and the information theory [30, 31, 40] based generalization bounds rely on the mutual information of the input (training data) and output (weights after training) of the learning algorithm. Another line is PAC Bayes [19] based method, which bounds the expectation of the error rates of a classifier chosen from a posterior distribution in terms of the KL divergence from a given prior distribution. Our research modifies the conventional Rademacher Complexity to calculate the complexity of the space explored by a learning algorithm, which in turn helps derive the generalization bound. Our approach resembles the first category, as we also rely on the complexity of the function space. However, our method differs as we focus on the function space explored by the learning trajectory, rather than the entire function space. The novelty of our technique lies in addressing the issue of dependence between training data and the function space explored by the learning trajectory, a dependence that is not permitted by the original Rademacher Complexity Theory.

**Generalization Analysis for SGD**   The optimization plays an nonnegligible role in the success of DNN. Therefore, there are many prior works studying the generalization of DNNs by exploring property of SGD, which could be summarized into two categories: stability of SGD and information-theoretic analysis. The most popular way of the former category is to analyze the stability of the weights updating. Hardt et al. [11] is the first work to analyze the stability of SGD with the requirements of smooth and Lipschitz assumptions. Its follow-up works try to discard the smooth [5], or Lipschitz [25] assumptions towards getting a more general bound. Information-theoretic methods leverage the chain rule of KL-divergence to calculate the mutual information between the learned model weights and the data. This kind of works is mainly applied for Stochastic Gradient Langevin Dynamics(SGLD), *i.e.* , SGD with noise injected in each step of parameters updating [28]. Negrea et al. [23], Haghifam et al. [10] improve the results using data-dependent priors. Neu et al. [24] construct an auxiliary iterative noisy process to adapt this method to the SGD scenario. In contrast to these studies, our approach utilizes more information related to learning trajectories. A more detailed comparison can be found in Table 2 and Appendix B.

## 3   Generalization Bound

Let us consider a supervised learning problem with a instance space $\mathcal{Z}$ and a parameter space $\mathcal{W}$. The loss function can be defined as $f : \mathcal{W} \times \mathcal{Z} \to \mathbb{R}_+$. We denote the distribution of the instance space $\mathcal{Z}$ as $\mu$. The $n$ i.i.d samples draw from $\mu$ are denoted as $S = \{z_1, ..., z_n\} \sim \mu^n$. Given parameters $\mathbf{w}$, the empirical risk and popular risk are denoted as $F_S(\mathbf{w}) \triangleq \frac{1}{n} \sum_i^n f(\mathbf{w}, z_i)$, and $F_\mu(\mathbf{w}) \triangleq \mathbb{E}_{z \sim \mu}[f(\mathbf{w}, z)]$ respectively. Our work studies the generalization error of the learned model, *i.e.* $F_\mu(\mathbf{w}) - F_S(\mathbf{w})$. For an optimizaiton process, the learning trajectory is represented as a function $\mathbf{J} : \mathbb{N} \to \mathcal{W}$. We use $\mathbf{J_t}$ to denote the weights of model after $t$ times updating, where $\mathbf{J_t} = \mathbf{J}(t)$. The learning algorithm is defined as $\mathcal{A} : \mu^n \times \mathbb{R} \to \mathbf{J}$, where the second input $\mathbb{R}$ denotes all randomness in the algorithm $\mathcal{A}$, including the randomness in initialization, batch sampling *et al.* . We simply use $\mathcal{A}(S)$ to represent a random choice for the second input term. Given two functions $U, V, \int_t U(t)\mathrm{d}V(t) \triangleq \sum_t U(t)(V(t+1) - V(t))$ and we use $\| \cdot \|$ to denote $L2$ norm. If $S$ is a set, then $|S|$ denotes the number of elements in $S$. $\mathbb{E}_t$ denotes taking the expectation conditioned on $\{\mathbf{J_i} | i \le t\}$.

Let mini-batch $B$ be a random subset sampled from dataset $S$, and we have $|B| = b$. The averaged function value of mini-batch $B$ is denoted as $F_B(\mathbf{w}) \triangleq \frac{1}{b} \sum_{z \in B} f(\mathbf{w}, z)$. The parameters updated with gradient descent can be formulated as:

$$\mathbf{J_{t+1}} = \mathbf{J_t} - \eta_t \nabla F_S(\mathbf{J_t}). \tag{1}$$

where $\eta_t$ is the learning rate for the $t$-th update. The parameter updating with stochastic gradient descent is:

$$\mathbf{J_{t+1}} = \mathbf{J_t} - \eta_t \nabla F_B(\mathbf{J_t}). \tag{2}$$

Let $\epsilon(\mathbf{w}) \triangleq \nabla F_S(\mathbf{w}) - \nabla F_B(\mathbf{w})$ be the gradient noise in mini-batch updating, where $\mathbf{w}$ is the weights of a DNN. Then we can transform Equation (2) into:

$$\mathbf{J_{t+1}} = \mathbf{J_t} - \eta_t \nabla F_S(\mathbf{J_t}) + \eta_t \epsilon(\mathbf{J_t}). \tag{3}$$

The covariance of the gradients over the entire dataset $S$ can be calculated as:

$$\Sigma(\mathbf{w}) \triangleq \frac{1}{n} \sum_{i=1}^n \nabla f(\mathbf{w}, z_i) \nabla f(\mathbf{w}, z_i)^{\mathrm{T}} - \nabla F_S(\mathbf{w}) \nabla F_S(\mathbf{w})^{\mathrm{T}}. \tag{4}$$

Therefore, the covariance of the gradient noise $\epsilon(\mathbf{w})$ is:

$$C(\mathbf{w}) \triangleq \frac{n - b}{b(n - 1)} \Sigma(\mathbf{w}). \tag{5}$$

Since for any $w$ we have $\mathbb{E}(\epsilon(\mathbf{w})) = 0$, we can represent $\epsilon(\mathbf{w})$ as $C(\mathbf{w})^{\frac{1}{2}} \epsilon'$, where $\epsilon'$ is a random distribution whose mean is zero and covariance matrix is an identity matrix. Here, $\epsilon'$ **can be any distributions**, including Guassian distribution [12] and $\mathcal{S}\alpha\mathcal{S}$ distribution [35].

The primary objective of our work is to suggest a new generalization bound that incorporates more comprehensive trajectory-related information. The **key aspects** of this information are: 1) It should be adaptive and change according to different learning trajectories. 2) It should not rely on the extra information from data distribution $\mu$ except from the training data $S$.

## 3.1 Investigating generalization alone learning trajectory

As annotated before, the learning trajectory is represented by a function $\mathbf{J} : \mathbb{N} \to \mathcal{W}$, which defines the relationship between the model weights and the training timesteps $t$. $\mathbf{J}_t$ denotes the model weights after $t$ times updating. Note that $\mathbf{J}$ depends on $S$, because it comes from the equation $\mathbf{J} = \mathcal{A}(S)$. We simply use $f(\mathbf{J_t}) : \mathcal{Z} \to \mathbb{R}_+$ to represent the function after $t$-times update. Our goal is to analyze the generalization error, i.e., $F_\mu(\mathbf{J_T}) - F_S(\mathbf{J_T})$, where $T$ represents the total training steps.

We reformulate the function corresponding to the finally obtained model as:

$$f(\mathbf{J_T}) = f(\mathbf{J_0}) + \sum_{t=1}^{T}(f(\mathbf{J_t}) - f(\mathbf{J_{t-1}})). \tag{6}$$

Therefore, the generalization error can be rewritten as:

$$F_\mu(\mathbf{J_T}) - F_S(\mathbf{J_T}) = \underbrace{F_\mu(\mathbf{J_0}) - F_S(\mathbf{J_0})}_{(i)} + \sum_{t=1}^{T}\underbrace{[(F_\mu(\mathbf{J_t}) - F_\mu(\mathbf{J_{t-1}})) - (F_S(\mathbf{J_t}) - F_S(\mathbf{J_{t-1}}))]}_{(ii)_t}. \tag{7}$$

In this form, we divide the generalization error into two parts. $(i)$ is the generalization error before the training. $(ii)_t$ is the generalization error caused by $t$-step update.

Typically, there is independence between $\mathbf{J_0}$ and the data $S$. Therefore, we have $\mathbb{E}(i) = 0$. Combining with this, we have:

$$\mathbb{E}[F_\mu(\mathbf{J_T}) - F_S(\mathbf{J_T})] = \mathbb{E}\sum_{t=1}^{T}(ii)_t. \tag{8}$$

Analyzing the generalization error after training can be transformed into analyzing the increase of generalization error for each update. This is a straighforward and quite different way to extract the information from learning trajectory compared with previous work. Here, we list two techniques that most used by previous works to extract the information from learning trajectory.

- (T1). This method leverages the chaining rule of mutual informaton to calculate a upper bound of the mutual information between $\mathbf{J_T}$ and the training data $S$, *i.e.* $I(S; \mathbf{J_T}) \le I(S; \mathbf{J_{t \le T}}) \le \sum_{t=0}^{T} I(S; \mathbf{J_t}|\mathbf{J_{i<t}})$. $I(S; \mathbf{J_T})$ is the value of concerning for their theory.
- (T2). This method assumes we have another data $S'$, which is obtained by replacing one sample in data $S$ with another sample drawing from distribution $\mu$. $\mathbf{J}'$ is the learning trajectory trained from data $S'$ with same randomness value as $\mathbf{J}$. Denote $\Delta_k \triangleq \|\mathbf{J_k} - \mathbf{J'_k}\|$ and assume $\Delta_0 = 0$. Then, the value of concerning is $\Delta_T$. The upper bound of $\Delta_T$ is calculate by iterately apply the formular $\Delta_k \le c_{k-1}\Delta_{k-1} + e_{k-1}$.

(T1) is commonly utilized in analyzing Stochastic Gradient Langevin Dynamics(SGLD) [18, 2, 28], while (T2) is frequently employed in stability-based works for analyzing SGD [11, 15, 5]. Our method offers several benefits, including: **1) We directly focus on the change in generalization error**, rather than intermediate values such as $\Delta_k$ and $I(S; \mathbf{J_t}|\mathbf{J_{i<t}})$, **2) The generalization error is equivalent to the sum of** $(ii)_t$, while (T1) and (T2) takes the upper bound value of $I(S; \mathbf{J_T})$ and $\Delta_T$, and **3) From this perspective, We can extract more in-depth trajectory-related information.** For (T1), the computation of $I(S; \mathbf{J_t}|\mathbf{J_{i<t}})$ primarily involves the information of $\nabla F_\mu(\mathbf{J_t})$, which is inaccessible to us (Detail in Appendix D and Neu et al. [24]). (T2) faces the challenge that only the upper bounds of $c_k$ and $e_k$ can be calculated. The upper bounds remain unchanged across various learning trajectories. Consequently, both (T1) and (T2) have difficulty conveying meaningful trajectory information.

## 3.2 A New Generalization Bound

In this section, we introduce the generalization bound based on our aforementioned modeling. Let us start with the definition of commonly used assumptions.

**Definition 3.1.** The function $f$ is $L$-Lipschitz, if for all $\mathbf{w_1}, \mathbf{w_2} \in \mathcal{W}$ and for all $z \in \mathcal{Z}$, wherein we have $\|f(\mathbf{w_1}, z) - f(\mathbf{w_2}, z)\| \le L\|\mathbf{w_1} - \mathbf{w_2}\|$.

**Definition 3.2.** The function $f$ is $\beta$-smooth, if for all $\mathbf{w_1}, \mathbf{w_2} \in \mathcal{W}$ and for all $z \in \mathcal{Z}$, wherein we have $\|\nabla f(\mathbf{w_1}, z) - \nabla f(\mathbf{w_2}, z)\| \leq \beta \|\mathbf{w_1} - \mathbf{w_2}\|$.

**Definition 3.3.** The function $f$ is convex, if for all $\mathbf{w_1}, \mathbf{w_2} \in \mathcal{W}$ and for all $z \in \mathcal{Z}$, wherein we have $f(\mathbf{w_1}, z) \geq f(\mathbf{w_2}, z) + (\mathbf{w_1} - \mathbf{w_2})^{\mathrm{T}} \nabla f(\mathbf{w_2}, z)$.

Here, $L$-lipschitz assumption implies that the $\|\nabla f(\mathbf{w}, z)\| \leq L$ holds. $\beta$-smooth assumption indicates the largest eignvalue of $\nabla^2 f(\mathbf{w}, z)$ is smaller than $\beta$. The convexity indicates the smallest eigenvalue of $\nabla^2 f(\mathbf{w}, z)$ are positive. These assumptions tell us the constraints of gradients and Hessian matrices of the training data and the unseen samples in the test set. Since the values of gradients and Hessian matrices in the training set are accessible, the key role of these assumptions is to deliver knowledge about the unseen samples in the test set.

In the following, we introduce a new generalization bound. We give the assumption required by our new generalization bound in the following.

**Assumption 3.4.** There is a value $\gamma$, so that for all $\mathbf{w} \in \{\mathbf{J_t} | t \in \mathbb{N}\}$, we have $\|\nabla F_\mu(\mathbf{w})\| \leq \gamma \|\nabla F_S(\mathbf{w})\|$.

*Remark* 3.5. Assumption 3.4 gives a restriction with the norm of popular gradient $\nabla F_\mu(\mathbf{w})$. This assumption is easily satisfied when $n$ is a large number, because we have $\lim_{n \to \infty} \|\nabla F_S(\mathbf{w})\| = \|\nabla F_\mu(\mathbf{w})\|$. When the $n$ is not large enough, the assumption will hold before SGD enter the neighbourhood of convergent point. Under the case that SGD enters the neighbourhood of convergent point, we give a relaxed assumption and its corresponding generalization bound in Appendix B. According to paper [42], this case will ununsually happen in real situation. Section 4 gives experiments to explore the assumption.

**Theorem 3.6.** *Under Assumption 3.4, given $S \sim \mu^n$, let $\mathbf{J} = \mathcal{A}(S)$, where $\mathcal{A}$ denoted the SGD or GD algorithm training with $T$ steps, we have:*

$$\mathbb{E}[F_\mu(\mathbf{J_T}) - F_S(\mathbf{J_T})] \leq -2\gamma' \mathbb{V}_m \mathbb{E} \int_t \frac{dF_S(\mathbf{J_t})}{\sqrt{n}} \sqrt{1 + \frac{\mathrm{Tr}(\Sigma(\mathbf{J_t}))}{\|\nabla F_S(\mathbf{J_t})\|_2^2}} + \mathcal{O}(\eta_m) \tag{9}$$

*where* $\mathbb{V}(\mathbf{w}) = \frac{\|\nabla F_S(\mathbf{w})\|}{\mathbb{E}_{U \subset S} \|\frac{|U|}{n} \nabla F_U(\mathbf{w}) - \frac{n-|U|}{n} \nabla F_{S/U}(\mathbf{w})\|}$, $\mathbb{V}_m = \max_t \mathbb{V}(\mathbf{J_t})$, $\gamma' = \max\{1, \max_{U \subset S; t} \frac{|U| \|\nabla F_U(\mathbf{J_t})\|}{n \|\nabla F_S(\mathbf{J_t})\|}\} \gamma$ *and* $\eta_m \triangleq \max_t \eta_t$.

*Remark* 3.7. Our generalization bound mainly relies on the information from gradients. $\mathbb{V}(\mathbf{w})$ is related to the variance of the gradient. When the variance of the gradients across different samples in the training set $S$ is large, then the value of $\mathbb{V}(\mathbf{w})$ is small, and vice versa. Note that we have $|U| < n$ due to $U \subset S$. Our bound will became trival if $\mathbb{E}_{U \subset S} \|\frac{|U|}{n} \nabla F_U(\mathbf{w}) - \frac{n-|U|}{n} \nabla F_{S/U}(\mathbf{w})\| = 0$. This rarely happens in real case, because it requires that for all $U \subset S$, we have $|U| \nabla F_U(\mathbf{w}) = (n - |U|) \nabla F_{S/U}(\mathbf{w})$. We also give a relaxed assumption version of this theorem in Appendix B. **The generalization bound provides a clear insight into how the reduction of training loss leads to a increase in generalization error.**

**Proof Sketch** The proof of this theorem is placed in Appendix A. Here, we give the sketch for this proof.

**Step 1** Beginning with Equation (8), we decomposite the $F_\mu(\mathbf{J_T}) - F_S(\mathbf{J_T})$ into a linear part $(\mathrm{gen}^{lin}(\mathbf{J_T}))$ and nonlinear part$(\mathrm{gen}^{nl}(\mathbf{J_T}))$. We have $\mathrm{gen}^{lin}(\mathbf{J_T}) = \sum_{t=1}^{T} (ii)_t^{lin}$, where $(ii)_t^{lin} \triangleq (\mathbf{J_t} - \mathbf{J_{t-1}})^{\mathrm{T}} (\nabla F_\mu(\mathbf{J_{t-1}}) - \nabla F_S(\mathbf{J_{t-1}}))$. The nonlinear part is $\mathrm{gen}^{nl}(\mathbf{J_T}) = F_\mu(\mathbf{J_T}) - F_S(\mathbf{J_T}) - \mathrm{gen}^{lin}(\mathbf{J_T})$. We takle these two parts differently. Here, we focus on analyzing $\mathrm{gen}^{lin}(\mathbf{J_T})$ because it dominates under small learning rate. Detail discussion of $\mathrm{gen}^{nl}(\mathbf{J_T})$ is given in Appendix (Propositon A.1 and Subsection C.3)

**Step 2** We construct the addictive linear space $\mathcal{L}_{\mathbf{J}|S} \triangleq \{\sum_{t=0}^{T-1} \mathbf{w_t}^{\mathrm{T}} \nabla f(\mathbf{J_t}) \mid \|\mathbf{w_t}\| \leq \Delta_t\}$, where $\Delta_t \triangleq \|\eta_t \nabla F_S(\mathbf{J_t})\|$. Then $\mathbb{E}[\mathrm{gen}^{lin}(\mathbf{J_T})] \leq 2\gamma' \mathbb{V}_m \mathbb{E} R_S(\mathcal{L}_{\mathbf{J}|S})$, where $R_S(\mathcal{L}_{\mathbf{J}|S}) \triangleq \mathbb{E}_\sigma \sup_{h \in \mathcal{L}_{\mathbf{J}|S}} (\frac{1}{n} \sum_{i=1}^{n} \sigma_i h(z_i))$.

Table 1: **Comparison of the generalization bounds with stability based method for SGD learning algorithms.** T.R.T is an abbreviation for the term related to trajectory. T.R.T is defined as the term that 1) varies based on different learning trajectories, and 2) don't rely on the extra information of data distribution $\mu$ except from training data $S$. We can infer that the proposed bound incorporates a greater amount of information pertaining to the trajectory. Other related works are discussed in Appendix D.

| Method | $\beta$-Smooth | $L$-Lipschitz | Convex | Small LR | Other Conditions | Generalization Bound | T.R.T |
|---|---|---|---|---|---|---|---|
| Hardt et al. [11] | ✓ | ✓ | ✓ | ✓ | | $\frac{2L^2}{n}\sum_{t=1}^T \eta_t$ | $\sum_{t=1}^T \eta_t$ |
| Hardt et al. [11] | ✓ | ✓ | | ✓ | $f\in[0,1], \eta_t < \frac{c}{t}$ | $O(\frac{1}{n}L^{\frac{2}{\beta c+1}}T^{\frac{\beta c}{\beta c+1}})$ | $T^{\frac{\beta c}{\beta c+1}}$ |
| Zhang et al. [43] | ✓ | ✓ | | ✓ | $T>n, \eta_t = \frac{c}{\beta t}$ | $\frac{16L^2 T^c}{n^{1+c}}$ | $T^c$ |
| Zhou et al. [44] | ✓ | ✓ | | ✓ | $\mathbb{E}_{z\in S}\|\nabla f(\mathbf{w},z)-\nabla F_S(\mathbf{w})\|^2 \le B^2$ | $O(\sqrt{\frac{1}{n}L\sqrt{2\beta F_\mu(\mathbf{J_0})}+\frac{1}{2}\mathbb{E}B^2\log T})$ | $\sqrt{\log T}$ |
| Bassily et al. [5] | | ✓ | ✓ | | Projected SGD | $2L^2\sqrt{\sum_{t=1}^{T-1}\eta_t^2}+\frac{4L^2}{n}\sum_{t=1}^{T-1}\eta_t$ | $\sum_{t=1}^{T-1}\eta_t$ |
| Lei and Ying [16] | | ✓ | ✓ | | Projected SGD | $O((1+\frac{T}{n^2})\sum_{t=1}^T \eta_t^2)$ | $T\sum_{t=1}^T \eta_t^2$ |
| Ours (Theorem 3.6) | | | | ✓ | $\|\nabla F_\mu(\mathbf{w})\|\le\gamma\|\nabla F_S(\mathbf{w})\|$ | Theorem 3.6 | $\int_t dF_S(\mathbf{J_t})\sqrt{1+\frac{Tr(\Sigma(\mathbf{J_t}))}{\|\nabla F_S(\mathbf{J_t})\|^2}}$ |

**Step 3** Finally, we compute the upper bound of $R_S(\mathcal{L}_{\mathbf{J}|S})$, which follows same techniques used in Radermacher Complexity theory. By combining this with Proposition A.1, we establish the theorem.

**Technical Novety** Directly applying the Rademacher complexity to calculate the generalization error bound fails because the large complexity of neural network's function space leads to trival bound[41]. In this work, we want to calculate the complexity of the function space that can be explored during the training process. However, there are two challenges here. **First**, the trajectory of neural network is a "line", instead of a function space that can be calculated the complexity. To solve this problem, we indroduce the addictive linear space $\mathcal{L}_{\mathbf{J}|S}$. This space contains the local information of learning trajectory, and can serve as the pseudo function space. **Second**, the function space $\mathcal{L}_{\mathbf{J}|S}$ has a dependent on the sample set $S$, while the theory of Rademacher complexity requires that the function space is independent with training samples. To decouple this dependence, we adapt the Rademacher complexity and we obtain that $\mathbb{E}[\text{gen}^{lin}(\mathbf{J_T})] \le 2\gamma'\mathbb{V}_m\mathbb{E}R_S(\mathcal{L}_{\mathbf{J}|S})$. Here, $\gamma'$ is indroduced to decouple the dependent fact mentioned above.

Next, in order to draw a clearer comparison with the stability-based method, we present the following corollary. This corollary employs the $\beta$-smooth assumption to bound $\text{gen}^{nl}(\mathbf{J_T})$ and leverages a similar learning rate setting to that found in stability based works.

**Corollary 3.8.** *If function $f(\cdot)$ is $\beta$-smooth, under Assumption 3.4 given $S \sim \mu^n$, let $\mathbf{J} = \mathcal{A}(S)$, $\eta_t = \frac{c}{\beta(t+1)}$, $M_2^2 = \max_t \mathbb{E}_{t-1}(\|\nabla F_S(\mathbf{J_t}) + \epsilon(\mathbf{J_t})\|^2)$ and $M_4^4 = \max_t \mathbb{E}_{t-1}(\|\nabla F_S(\mathbf{J_t}) + \epsilon(\mathbf{J_t})\|^4)$ , where $\mathcal{A}$ denoted the SGD or GD algorithm training with $T$ steps, we have:*

$$
\mathbb{E}[F_\mu(\mathbf{J_T}) - F_S(\mathbf{J_T})] \le - 2\gamma'\mathbb{V}_m\mathbb{E}\int_t \frac{dF_S(\mathbf{J_t})}{\sqrt{n}}\sqrt{1+\frac{\text{Tr}(\Sigma(\mathbf{J_t}))}{\|\nabla F_S(\mathbf{J_t})\|_2^2}}
$$
$$
+ 2c^2\gamma'\mathbb{V}_m M_4^2\sqrt{\mathbb{E}\int_t \frac{dt}{n\beta^2(t+1)^4}\left(1+\frac{\text{Tr}(\Sigma(\mathbf{J_t}))}{\|\nabla F_S(\mathbf{J_t})\|_2^2}\right)} \quad (10)
$$
$$
+ 2c^2\frac{M_2^2}{\beta}.
$$

*where* $\mathbb{V}(\mathbf{w}) = \frac{\|\nabla F_S(\mathbf{w})\|}{\mathbb{E}_{U\subset S}\|\frac{|U|}{n}\nabla F_U(\mathbf{w}) - \frac{n-|U|}{n}\nabla F_{S/U}(\mathbf{w})\|}$, $\mathbb{V}_m = \max_t \mathbb{V}(\mathbf{J_t})$ *and* $\gamma' = \max\{1, \max_{U\subset S;t} \frac{|U|\|\nabla F_U(\mathbf{J_t})\|}{n\|\nabla F_S(\mathbf{J_t})\|}\}\gamma$.

### 3.3 Further Analysis

#### 3.3.1 Interpreting the Generalization Bounds

We rewrite the obtained generalization bound here:

$$\mathbb{E}[F_\mu(\mathbf{J_T}) - F_S(\mathbf{J_T})] \leq \underbrace{\gamma'}_{\text{Bias of Training Set}} \overbrace{\mathbb{V}_m}^{\frac{1}{\text{Diversity of Training Set}}} \underbrace{\left(-2\mathbb{E}\int_t \frac{dF_S(\mathbf{J_t})}{\sqrt{n}}\sqrt{1 + \frac{\text{Tr}(\Sigma(\mathbf{J_t}))}{\|\nabla F_S(\mathbf{J_t})\|_2^2}}\right)}_{\text{Complexity of Learning Trajectory}} + \mathcal{O}(\eta_m)$$

$$(11)$$

The "Bias of Training Set" refers to the disparity between the characteristics of the training set and those of the broader population. To measure this difference, we use the distance between the norm of the popular gradient and that of the training set gradient, as specified in Assumption 3.4. The "Diversity of Training Set" can be understood as the variation among the samples in the training set, which in turn affects the quality of the training data. The ratio $\frac{\text{Bias of Training Set}}{\text{Diversity of Training Set}}$ gives us the property of information conveyed by the training set. It is important to consider the properties of the training set, as the data may not contribute equally to the generalization[36]. **The detail version of the equation can be found in Theorem 3.6.**

#### 3.3.2 Asymptotic Analysis

We will first analyze the dependent of $n$ for $\mathbb{V}$. The $\mathbb{V}$ is calculated as $\mathbb{V}(\mathbf{w}) = \frac{\|\nabla F_S(\mathbf{w})\|}{\mathbb{E}_{U \subset S}\|\frac{|U|}{n}\nabla F_U(\mathbf{w}) - \frac{n-|U|}{n}\nabla F_{S/U}(\mathbf{w})\|}$. Obviously, the gradient of individual sample is unrelated to the sample size $n$. And $|U| \sim n$. Therefore, $\mathbb{V} = \mathcal{O}(1)$. Similarly, we have $\mathbb{E}\int_t \frac{dF_S(\mathbf{J_t})}{\sqrt{n}}\sqrt{1 + \frac{\text{Tr}(\Sigma(\mathbf{J_t}))}{\|\nabla F_S(\mathbf{J_t})\|_2^2}} = \mathcal{O}(\frac{1}{\sqrt{n}})$. As for the $\mathcal{O}(\eta_m)$ term in Theorem 3.6, we have $\lim_{n\to\infty} \mathcal{O}(\eta_m) = 0$ according to Proposition A.1. We simply assume that $\mathcal{O}(\eta_m) = \mathcal{O}(\frac{1}{n^c})$. Therefore, our bound has $\mathcal{O}(\frac{1}{n^{\min\{0.5, c\}}})$.

#### 3.3.3 Comparison with Stability-based methods

**We first compare our method with the stability-based methods in terms of the trajectory information.** In Table 1, we present a summary of stability-based methods, while other methods are outlined in Appendix D. We focus on generalization bounds from previous works that eliminate terms dependent on extra information about data distribution $\mu$, apart from the training data $S$, using assumptions such as smoothness or Lipschitz continuity. Analyzing Table 1 reveals that most prior works primarily depend on the learning rate $\eta$ and the total number of training steps $T$. This suggests that we can achieve the same bound by using an identical learning rate schedule and total training steps, which does not align with our practical experience. Our proposed generalization bound considers the evolution of function values, gradient covariance, and gradient norms throughout the training process. As a result, our bounds encompass more comprehensive information about the learning trajectory.

Table 2: **Detail comparison with Hardt et al. [11].** The $\beta$ refers to the $\beta$-smooth assumption (see in Definition 3.2). $S$ denotes the training set.

| | Uniform Stability[11] | Ours |
|---|---|---|
| Assumption | $\forall \mathbf{w} \in \mathcal{W} \quad \forall z' \in \mathcal{Z} \quad \|\nabla f(\mathbf{w}, z')\| \leq L$ | $\forall \mathbf{w} \in \{\mathbf{J_t} \mid t \in \mathbb{N}\} \quad \|\mathbb{E}_{z' \sim \mu}\nabla f(\mathbf{w}, z')\| \leq \gamma\|\mathbb{E}_{z \in S}\nabla f(\mathbf{w}, z')\|$ |
| Modelling Method of SGD | Epoch Structure | Full Batch Gradient + Stochastic Noise |
| Batch Size | 1 | $\leq n$ |
| Trajectory Information in Bound | Learning rate and number of training step | Values in Trajectory (gradient norm and covariance) |
| Perspective | Stability of Algorithm | Complexity of Learning Trajectory |

**Next, we give a detail comparison with Hardt et al. [11] in Table 2.** The concept of uniform stability is commonly used to evaluate the ability of SGD in generalizaton, by assessing its stability when a single training sample is altered. Our primary point of comparison is with Hardt et al. [11], as their work is considered the most representative in terms of analyzing the stability of SGD. We find that **First**, the assumption of Uniform Stability requires the gradient norm of all input samples for all weights being bounded by $L$, whereas our assumption only limits the expectation of the gradients

for the weights during the learning trajectory. **Secondly**, Uniform Stability uses an epoch structure to model the stochastic gradient descent, whereas our approach regards each stochastic gradient descent as full batch gradient descent with added stochastic noise. The epoch structure complicates the modelling process because it requires a consideration of sampling. As a result, in Hardt et al. [11], the author only considers the setting with batch size 1. **Thirdly**, the bound of Uniform Stability only uses hyperparameters setting such as learning rate and number of training step. In contrast, our bound contains more trajectory-related information, such as the gradient norm and covariance. **Finally**, the Uniform Stability provides the generalization bound based on the stability of the algorithm, while our approach leverages the complexity of the learning trajectory. **In summary**, there are some notable differences between our approach and Uniform Stability, such as the assumptions made, the modelling process, the type of information used in the bound, and the perspectives.

## 4 Experiments

### 4.1 Tightness of Our Bounds

Table 3: **Numeric comparison with stability-based work on toy examples.** The reason for the value of Zhang et al. [43] is large is because that our and Hardt et al. [11] has dependent on $\frac{L^2}{\beta}$, while Zhang et al. [43] depends on $L^2$. $L$ and $\beta$ are usually large numbers.

| Gen Error | Ours | Hardt et al. [11] | Zhang et al. [43] |
|-----------|------|-------------------|-------------------|
| 1.49 | 3.62 | 4.04 | 4417 |

In a toy dataset setting, we compare our generalization bound with stability-based methods.

**Reasons for toy examples** **1) Some values in the bounds are hard to be calculated.** Calculating $\beta$ (under the $\beta$-smooth assumption) and $L$ (under the $L$-Lipschitz assumption) in stability-based work, as well as the values of $\mathbb{V}$ and $\gamma$ in our proposed bound, are challenging. **2) Stability-based methods require a batch size of 1.** The training is hard for batch size of 1 with learning rate setting $\eta_t = \frac{1}{t}$ in complex datasets.

**Constuction of the toy examples** In the following, we discuss the construction of the toy dataset used to compare the tightness of the generalization bounds. The training data is $X_{tr} = \{x_i\}_{i=1}^n$. All the data $x_i$ is sampled from Guassian distribution $\mathcal{N}(0, \mathbf{I}_d)$. Sampling $\tilde{\mathbf{w}} \sim \mathcal{N}(0, \mathbf{I}_d)$, the ground truth is generated by $y_i = 1$ if $\tilde{\mathbf{w}}^{\mathrm{T}} x_i > 0$ else $0$. The weights for learning is denoted as $\mathbf{w}$. The predict $\tilde{y}$ is calculated as $\tilde{y}_i = \mathbf{w}^{\mathrm{T}} x_i$. The loss for a simple data point is $l_i = \|y_i - \mathbf{w}^{\mathrm{T}} x_i\|_2$. The training loss is $\mathcal{L} = \sum_{i=1}^n l_i$. The test data is $X_{te} = \{x_i'\}$, where $x_i' = \tilde{x}_i'$ and $\tilde{x}_i' \sim \mathcal{N}(0, \mathbf{I}_d)$. We use 100 samples for training and 1,000 samples for evaluation. The model is trained using SGD for 200 epochs.

We evaluate the tightness of our bound by comparing our results with those in Hardt et al. [11] and Zhang et al. [43] from the original paper. We set the learning rate as $\eta_t = \frac{1}{\beta t}$. **Our reasons for comparing with these two papers are**: 1) Hardt et al. [11] is a representative study, 2) Both papers have theorems using a learning rate setting of $\eta_t = \mathcal{O}(\frac{1}{t})$, which aligns with Corollary 3.8 in our paper, and 3) They do not assume convexity. The generalization bounds we compare include Corollary 3.8 from our paper, Theorem 3.12 from Hardt et al. [11], and Theorem 5 from Zhang et al. [43].

Our results are given in Table 3. Our bound is tighter under this setting.

### 4.2 Capturing the trend of generalization error

In this section, 1) we conduct the deep learning experiment to verify Assumption 3.4 and 2) Verify whether our proposed generalization bound can capture the changes of generalization error. In this experiment, we mainly consider the term $\mathcal{C}(\mathbf{J_t}) \triangleq -2 \int_{i=0}^t \frac{dF_S(\mathbf{J_i})}{\sqrt{n}} \sqrt{1 + \frac{\mathrm{Tr}(\Sigma(\mathbf{J_i}))}{\|\nabla F_S(\mathbf{J_i})\|_2^2}}$. We omit the term $\gamma'$ and $\mathbb{V}_m$, because all the trajectory related information that we want to explore is stored in $\mathcal{C}(\mathbf{J_t})$. Capturing the trend of generalization error is regarded as an important problem in Nagarajan [21]. Unless further specified, we use the default setting of the experiments on CIFAR-10 dataset [14]

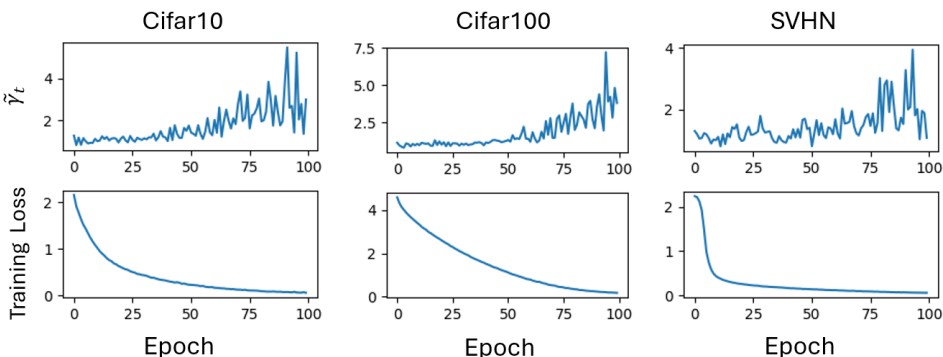

Figure 1: **Exploration of Assumption 3.4 for different dataset.** The $\widetilde{\gamma}_t$ is stable before training loss reaches a relative small value. Assumption holds if the training is stop before extremely overfitting. A relaxed assumption and its corresponding generalization bound are given in Appendix B for extremely overfitting situation

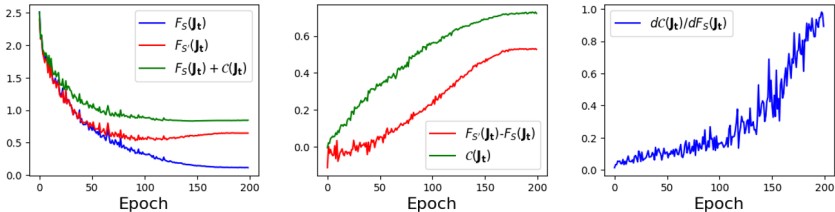

Figure 2: **Exploration of $\mathcal{C}(\mathbf{J_t})$ during the training process.** *Left:* The curve $F_S(\mathbf{J_t}) + \mathcal{C}(\mathbf{J_t})$ exhibits a comparable Pattern with the curve $F_{S'}(\mathbf{J_t})$. *Center:* After the early stage, $\mathcal{C}(\mathbf{J_t})$ and $\nabla F_{S'}(\mathbf{J_t}) - \nabla F_S(\mathbf{J_t})$ have a similar trend. *Right:* The value of $\frac{d\mathcal{C}(\mathbf{J_t})}{dF_S(\mathbf{J_t})}$ alone the training process.

with the VGG13 [34] network. The experimental details for each figure can be found in Appendix C.2.

Our observations are:

- Assumption 3.4 is valid when SGD is not exhibiting extreme overfitting.
- The term of $\mathcal{C}(\mathbf{J_t})$ can depict how the generalization error varies along the training process. And it can also track the changes in generalization error when adjustments are made to learnling rates and label noise levels

**Exploring the assumption 3.4 for different dataset during the training process** To explore the Assumption 3.4, we define $\gamma_t \triangleq \frac{\|\nabla F_\mu(\mathbf{J_t})\|}{\|\nabla F_S(\mathbf{J_t})\|}$ and $\widetilde{\gamma}_t \triangleq \frac{\|\nabla F_{S'}(\mathbf{J_t})\|}{\|\nabla F_S(\mathbf{J_t})\|}$, where $S'$ is another data set i.i.d sampled from distribution $\mu$. Because $S'$ is independent with $S$, we have $\widetilde{\gamma}_t \approx \gamma_t$. We found that $\widetilde{\gamma}_t$ is stable around 1 during the early stage of training(Figure 1). When the training loss is reaching a relative small value, $\widetilde{\gamma}_t$ increases as we continue training. This phenomenon remain consistant aross the Cifar10, Cifar100 and SVHN datasets. The $\gamma$ in Assumption 3.4 can be assigned as $\gamma = \max_t \widetilde{\gamma}_t$. We can always find such $\gamma$ if the optimizer is not extreme overfitting. Under the extremely overfitting case, we can use the relaxed theorem in Appendix B to bound the generalization error.

**The bound capturing the trend of generalization error during training process** The generalization error and the $\mathcal{C}(\mathbf{J_t})$ both changes as the training continues. Therefore, we want to verify whether they correlate with each other during the training process. Here, we use the term $\nabla F_{S'}(\mathbf{J_t}) - \nabla F_S(\mathbf{J_t})$ to approximate the generalization error. We find that $\nabla F_{S'}(\mathbf{J_t}) - \nabla F_S(\mathbf{J_t})$ has similar trend with $\mathcal{C}(\mathbf{J_t})$ (Figure 2 *Center*). What's more, we also find that the curve of $\nabla F_S(\mathbf{J_t}) + \mathcal{C}(\mathbf{J_t})$ exhibits a comparable pattern with the curve $F_{S'}(\mathbf{J_t})$ (Figure 2 *Left*). To explore whether $\mathcal{C}(\mathbf{J_t})$ reveals influence of the change of $F_S(\mathbf{J_t})$ to the generalization error, we plot $\frac{d\mathcal{C}(\mathbf{J_t})}{dF_S(\mathbf{J_t})}$ (Figure 2 *Right*) during

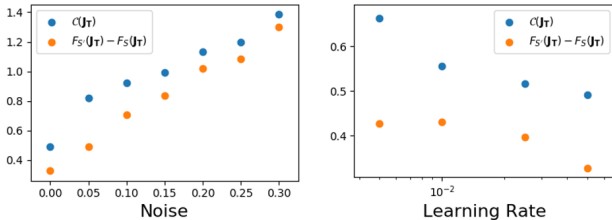

Figure 3: $\mathcal{C}(\mathbf{J_T})$ **correlates with** $\nabla F_{S'}(\mathbf{J_t}) - \nabla F_S(\mathbf{J_t})$. *Left:* $\mathcal{C}(\mathbf{J_t})$ and the generalization error under different label noise level. *Right:* $\mathcal{C}(\mathbf{J_t})$ and the generalization error under learning rate. The $\mathcal{C}(\mathbf{J_t})$ can capture the trend of generalization error cased by learning rate when learning rate is small. Appendix E provides proof that a large learning rate results in a smaller proposed generalization bound. Further discussions on why a small learning rate leads to a larger generalization error can be found in Li et al. [17], Barrett and Dherin [3].

the training process. $\frac{d\mathcal{C}(\mathbf{J_t})}{dF_S(\mathbf{J_t})}$ increases slowly during the early stage of training, but surge rapidly afterward. This discovery is aligned with our intuition about the overfitting.

**The complexity of learning trajectory correlates with the generalization error** In Figure 3, we carry out experiments under various settings. Each data point in the figure represents the average of three repeated experiments. The results demonstrate that both the generalization error and $\mathcal{C}(\mathbf{J_t})$ increase as the level of label noise is raised (Figure 3 *Left*). The another experiments measure $\mathcal{C}(\mathbf{J_t})$ and generalization error for different learning rate and discover that $\mathcal{C}(\mathbf{J_t})$ can capture the trend generalization error. The reasons behind a larger learning rate resulting in a smaller generalization error have been explored in Li et al. [17], Barrett and Dherin [3]. Additionally, Appendix E discusses why a larger learning rate can lead to a smaller $\mathcal{C}(\mathbf{J_t})$.

## 5   Limitation

The assumption of small learning rate is required by our method. But this assumption is also common use in previous works. For example, Hardt et al. [11], Zhang et al. [43], Zhou et al. [44] explicitly requires that the learning rate should be small and is decayed with a rate of $\mathcal{O}(\frac{1}{t})$. Some methods have no explict requirements about this but show that large learning rate pushes the generalization bounds to a trivial point. For example, the generalization bounds in works [5, 16] have a term $\sum_{t=1}^{T} \eta_t^2$ that is not decayed as the data size $n$ increases. The value of this term is unignorable when the learning rate is large. The small learning assumption widens the gap between theory and practice. Eliminating this assumption is crucial for future work.

## 6   Conclusion

In this study, we investigate the relation between learning trajectories and generalization capabilities of Deep Neural Networks (DNNs) from a unique standpoint. We show that learning trajectories can serve as reliable predictors for DNNs' generalization performance. To understand the relation between learning trajectory and generalization error, we analyze how each update step impacts the generalization error. Based on this, we propose a novel generalization bound that encompasses extensive information related to the learning trajectory. The conducted experiments validate our newly proposed assumption. Experimental findings reveal that our method effectively captures the generalization error throughout the training process. Furthermore, our approach can also track changes in generalization error when adjustments are made to learning rates and the level of label noises.

## 7   Acknowledgement

We thank all the anonymous reviewers for their valuable comments. The work was supported in part with the National Natural Science Foundation of China (Grant No. 62088102).

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

# A Proof of Theorem 3.6

We rewrite the Equation (7) and Equation (8):

$$F_\mu(\mathbf{J_T}) - F_S(\mathbf{J_T}) = \underbrace{F_\mu(\mathbf{J_0}) - F_S(\mathbf{J_0})}_{(i)} + \sum_{t=1}^{T} \underbrace{[(F_\mu(\mathbf{J_t}) - F_\mu(\mathbf{J_{t-1}})) - (F_S(\mathbf{J_t}) - F_S(\mathbf{J_{t-1}}))]}_{(ii)_t},$$

(12)

and

$$\mathbb{E}[F_\mu(\mathbf{J_T}) - F_S(\mathbf{J_T})] = \mathbb{E}\sum_{t=1}^{T}(ii)_t.$$

(13)

Using Taylor expansion for the function $f(\cdot)$, we have:

$$f(\mathbf{J_t}) - f(\mathbf{J_{t-1}}) = (\mathbf{J_t} - \mathbf{J_{t-1}})^{\mathrm{T}}\nabla f(\mathbf{J_{t-1}}) + \mathcal{O}(\|\mathbf{J_{t+1}} - \mathbf{J_t}\|^2).$$

(14)

Therefore, we can define $(ii)_t^{lin}$ as:

$$(ii)_t^{lin} \triangleq (\mathbf{J_t} - \mathbf{J_{t-1}})^{\mathrm{T}}(\nabla F_\mu(\mathbf{J_{t-1}}) - \nabla F_S(\mathbf{J_{t-1}})).$$

(15)

The $(ii)_t$ can be decomposed as $(ii)_t = (ii)_t^{lin} + (ii)_t^{nl}$, where $(ii)_t^{nl} \triangleq (ii)_t - (ii)_t^{lin}$.

Then Equation 13 can be decomposited as:

$$\mathbb{E}[F_\mu(\mathbf{J_T}) - F_S(\mathbf{J_T})] = \mathbb{E}\underbrace{\sum_{t=1}^{T}(ii)_t^{lin}}_{\mathrm{gen}^{lin}(\mathbf{J_T})} + \mathbb{E}\underbrace{\sum_{t=1}^{T}(ii)_t^{nl}}_{\mathrm{gen}^{nl}(\mathbf{J_T})}.$$

(16)

**Proposition A.1.** *For the gradient descent or the stochastic gradient descent algorithm, we have:*

$$\mathbb{E}[\mathrm{gen}^{lin}(\mathbf{J_T})] = \mathbb{E}[\sum_{t=0}^{T-1}\eta_t\nabla F_S(\mathbf{J_t})^{\mathrm{T}}(\nabla F_S(\mathbf{J_t}) - \nabla F_\mu(\mathbf{J_t}))].$$

(17)

*If $T = \mathcal{O}(\frac{1}{\eta_m})$, we have:*

$$|\mathrm{gen}^{nl}(\mathbf{J_T})| = \mathcal{O}(\eta_m),$$

(18)

*where $\eta_m \triangleq \max_t \eta_t$, and we have:*

$$\lim_{n\to\infty}|\mathrm{gen}^{nl}(\mathbf{J_T})| = 0.$$

(19)

*Remark* A.2. We give an experimental exploration of the $\mathrm{gen}^{nl}(\mathbf{J_T})$ in Appendix C.3. We discover that if the optimizer doesn't enter the EoS (Edge of Stability) regime [9], we have $\mathrm{gen}^{nl}(\mathbf{J_T}) \approx 0$.

*Proof.* **Analyzing of** $\mathrm{gen}^{lin}(\mathbf{J_T})$

Because of $\epsilon(\mathbf{w}) = C(\mathbf{w})^{\frac{1}{2}}\epsilon'$ and $\mathbb{E}[\epsilon'] = 0$ (detail in Equation (3) and Equation (5)d), we can get:

$$\begin{aligned}
&\mathbb{E}_{t-1}[\epsilon_t^{\mathrm{T}}(\nabla F_\mu(\mathbf{J_t}) - \nabla F_S(\mathbf{J_t}))] \\
&= \mathbb{E}_{t-1}[(\epsilon')^{\mathrm{T}}(C(\mathbf{J_t})^{\frac{1}{2}})^{\mathrm{T}}(\nabla F_\mu(\mathbf{J_t}) - \nabla F_S(\mathbf{J_t}))] \\
&= \mathbb{E}_{t-1}[\epsilon']^{\mathrm{T}}\mathbb{E}_{t-1}[(C(\mathbf{J_t})^{\frac{1}{2}})^{\mathrm{T}}(\nabla F_\mu(\mathbf{J_t}) - \nabla F_S(\mathbf{J_t}))] \\
&= 0.
\end{aligned}$$

(20)

Combining with Formula (3), we have

$$\mathbb{E}[\mathrm{gen}^{lin}(\mathbf{J_T})] = \mathbb{E}[\sum_{t=0}^{T-1}\eta_t\nabla F_S(\mathbf{J_t})^{\mathrm{T}}(\nabla F_S(\mathbf{J_t}) - \nabla F_\mu(\mathbf{J_t}))].$$

(21)

**Analyzing of** $\text{gen}^{nl}(\mathbf{J_T})$

Here, we denote $M \triangleq \max_t \|\nabla F_S(\mathbf{J_t})\|$. According to the definition of $\text{gen}^{nl}(\mathbf{J_T})$.

$$
\begin{aligned}
|\text{gen}^{nl}(\mathbf{J_T})| &\leq |F_\mu(\mathbf{J_T}) - F_\mu^{lin}(\mathbf{J_T})| + |F_S(\mathbf{J_T}) - F_S^{lin}(\mathbf{J_T})| \\
&= |\sum_{t=1}^{T} \mathcal{O}(\|\mathbf{J_{t+1}} - \mathbf{J_t}\|^2)| + |\sum_{t=1}^{T} \mathcal{O}(\|\mathbf{J_{t+1}} - \mathbf{J_t}\|^2)| \\
&= \sum_{t=1}^{T} \mathcal{O}(\eta_t^2 \|\nabla F_S(\mathbf{J_t})\|^2) \\
&= \mathcal{O}(T\eta_m^2 M^2) \\
&= \mathcal{O}(\frac{1}{\eta_m} \eta_m^2 M^2) \\
&= \mathcal{O}(\eta_m)
\end{aligned}
\tag{22}
$$

Because all the elements of training set $S$ are sampled from distribution $\mu$, we have $\lim_{n\to\infty} \nabla F_S(\mathbf{w}) = F_\mu(\mathbf{w})$, Therefore:

$$
\lim_{n\to\infty} (ii)_t^{lin} = \lim_{n\to\infty} (\mathbf{J_t} - \mathbf{J_{t-1}})^{\mathrm{T}} (\nabla F_\mu(\mathbf{J_{t-1}}) - \nabla F_S(\mathbf{J_{t-1}})) = 0.
\tag{23}
$$

What's more, we also have:

$$
\lim_{n\to\infty} [F_\mu(\mathbf{J_T}) - F_S(\mathbf{J_T})] = 0.
\tag{24}
$$

Because $F_\mu(\mathbf{J_T}) - F_S(\mathbf{J_T}) = \sum_{t=1}^{T} (ii)_t^{lin} + \sum_{t=1}^{T} (ii)_t^{nl}$, we have:

$$
\lim_{n\to\infty} |\text{gen}^{nl}(\mathbf{J_T})| = \lim_{n\to\infty} \left| \sum_{t=1}^{T} (ii)_t^{nl} \right| = \lim_{n\to\infty} \left| F_\mu(\mathbf{J_T}) - F_S(\mathbf{J_T}) - \sum_{t=1}^{T} (ii)_t^{lin} \right| = |0 - 0| = 0
\tag{25}
$$

$\square$

According to the Equation (17), we analyze the generalization error of $\mathcal{F}_{\mathbf{J}|S} \triangleq \{\sum_{t=0}^{T-1} \mathbf{w_t}^{\mathrm{T}} \nabla f(\mathbf{J_t}) \mid \mathbf{w_t} = \delta_t \frac{\nabla f(\mathbf{J_t})}{\|\nabla f(\mathbf{J_t})\|}\}$ as a proxy for analyzing generalization error of the function trained using SGD or GD algorithm. The value of $\text{gen}^{lin}(\mathbf{J_T})$ is equal to the generalization error of $\mathcal{F}_{\mathbf{J}|S}$. To analyze $\mathcal{F}_{\mathbf{J}|S}$, we construct an addictive linear space as $\mathcal{L}_{\mathbf{J}|S} \triangleq \{\sum_{t=0}^{T-1} \mathbf{w_t}^{\mathrm{T}} \nabla f(\mathbf{J_t}) \mid \|\mathbf{w_t}\| \leq \delta_t\}$, where $\delta_t \triangleq \|\eta_t \nabla F_S(\mathbf{J_t})\|$. Here, we use $\mathbf{J}|S$ to emphasize that $\mathbf{J}$ depends on $S$.

Under Assumption 3.4 (that is introduced in the main paper), we can have the following lemma.

**Lemma A.3.** *Under Assumption 3.4, given $S \sim \mu^n$, let $\mathbf{J} = \mathcal{A}(S)$, we have:*

$$
\mathbb{E}[\text{gen}^{lin}(\mathbf{J_T})] \leq 2\gamma' \mathbb{V}_m \mathbb{E} R_S(\mathcal{L}_{\mathbf{J}|S}),
\tag{26}
$$

*where* $\mathbb{V}(\mathbf{w}) = \frac{\|\nabla F_S(\mathbf{w})\|}{\mathbb{E}_{U \subset S} \| \frac{|U|}{n} \nabla F_U(\mathbf{w}) - \frac{|S|-|U|}{n} \nabla F_{S/U}(\mathbf{w})\|}$, $\mathbb{V}_m = \max_t \mathbb{V}(\mathbf{J_t})$ *and* $\gamma' = \max\{1, \max_{U \subset S; t} \frac{|U| \|\nabla F_U(\mathbf{J_t})\|}{n \|\nabla F_S(\mathbf{J_t})\|}\} \gamma$.

*Proof.* For a function $h$, we define that $h_\mu \triangleq \mathbb{E}_{z \sim \mu}[g(z)]$ and $h_S = \frac{1}{n} \sum_{z_i \in S} h(z_i)$. Given a function space, the maximum generalization error of the space can be defined as: $\Phi(S, H) \triangleq \sup_{h \in H} (h_\mu - h_S)$

$$
\begin{aligned}
\Phi(S, H|S) &= \sup_{h \in H|S} (h_\mu - h_S) \\
&= \sup_{h \in H|S} (\mathbb{E}_{S'} h_{S'} - h_S) \\
&\leq \mathbb{E}_{S'} \sup_{h \in H|S} (h_{S'} - h_S) \\
&= \mathbb{E}_{S',\sigma} \sup_{h \in H|S} (\frac{1}{n} \sum_i^n \sigma_i (h(z_i^{\sigma_i}) - h(z_i^{-\sigma_i}))) \\
&\leq \mathbb{E}_{S',\sigma} \sup_{h \in H|S} (\frac{1}{n} \sum_i^n \sigma_i (h(z_i^{\sigma_i}))) + \mathbb{E}_{S',\sigma} \sup_{h \in H|S} (\frac{1}{n} \sum_i^n \sigma_i (h(z_i^{\sigma_i}))) \\
&= 2\mathbb{E}_{S',\sigma} \sup_{h \in H|S} (\frac{1}{n} \sum_i^n \sigma_i (h(z_i^{\sigma_i}))),
\end{aligned}
\tag{27}
$$

where $S'$ is another i.i.d sample set drawn from $\mu^n$ and $\sigma$ denotes the Rademacher variable. The $\sigma_i$ in $z_i^{\sigma_i}$ denotes $z_i^{\sigma_i}$ that belongs to $S$ or $S'$. if $\sigma_i = -1$ $z_i^{\sigma_i} \in S$, otherwise, $z_i^{\sigma_i} \in S'$.

$$
\begin{aligned}
R_S(\mathcal{L}_{\mathbf{J}|S}) &\triangleq \mathbb{E}_\sigma \sup_{h \in \mathcal{L}_{\mathbf{J}|S}} (\frac{1}{n} \sum_i^n \sigma_i h(z_i)) \\
&= \mathbb{E}_\sigma \sup_{h \in \mathcal{L}_{\mathbf{J}|S}} (\frac{1}{n} (\sum_{z \in S_+} h(z) - \sum_{z \in S_-} h(z))) \\
&= \mathbb{E}_\sigma (\frac{1}{n} \sum_{t=0}^{\mathrm{T}} \delta_t \| g_{S_+}(\mathbf{J_t}) - g_{S_-}(\mathbf{J_t}) \|),
\end{aligned}
\tag{28}
$$

where $S_+ \triangleq \{z_i \mid \sigma_i = +1\}$ and $S_- \triangleq \{z_i \mid \sigma_i = -1\}$, and $g_S(\mathbf{w}) \triangleq |S| \nabla F_S(\mathbf{w})$.

$$
\begin{aligned}
\mathbb{E}_{S',\sigma} \sup_{h \in \mathcal{F}_{\mathbf{J}|S}} (\frac{1}{n} \sum_i^n \sigma_i (h(z_i^{\sigma_i}))) &= \mathbb{E}_{S',\sigma} \sup_{h \in \mathcal{F}_{\mathbf{J}|S}} (\frac{1}{n} (\sum_{z \in S'_+} h(z) - \sum_{z \in S_-} h(z))) \\
&= \mathbb{E}_{S',\sigma} (\frac{1}{n} \sum_{t=0}^{T-1} \delta_t \frac{g_S(\mathbf{J_t})}{\| g_S(\mathbf{J_t}) \|} (g_{S'_+}(\mathbf{J_t}) - g_{S_-}(\mathbf{J_t}))) \\
&= \mathbb{E}_{S',\sigma} (\frac{1}{n} \sum_{t=0}^{T-1} \delta_t \frac{g_S(\mathbf{J_t})}{\| g_S(\mathbf{J_t}) \|} (|S_+| \nabla F_\mu(\mathbf{J_t}) - g_{S_-}(\mathbf{J_t})))
\end{aligned}
\tag{29}
$$

where $S'_+$ is a subset of $S'$ with $|S'_+| = |S_+|$. Defining $k \triangleq \gamma' \mathbb{V}_m$, we have:

$$k\mathbb{E}_\sigma \sup_{h\in\mathcal{L}_{\mathbf{J}|S}} (\frac{1}{n}\sum_i^n \sigma_i h(z_i)) - \mathbb{E}_{S',\sigma} \sup_{h\in\mathcal{F}_{\mathbf{J}|S}} (\frac{1}{n}\sum_i^n \sigma_i(h(z_i)))$$

$$= k\mathbb{E}_\sigma(\frac{1}{n}\sum_{t=0}^T \delta_t \|g_{S_+}(\mathbf{J_t}) - g_{S_-}(\mathbf{J_t})\|) - \mathbb{E}_{S',\sigma}(\frac{1}{n}\sum_{t=0}^T \delta_t \frac{g_S(\mathbf{J_t})}{\|g_S(\mathbf{J_t})\|}(g_{S'_+}(\mathbf{J_t}) - g_{S_-}(\mathbf{J_t})))$$

$$= k\mathbb{E}_\sigma(\frac{1}{n}\sum_{t=0}^T \delta_t \|g_{S_+}(\mathbf{J_t}) - g_{S_-}(\mathbf{J_t})\| - \frac{1}{n}\sum_{t=0}^T \delta_t \frac{g_S(\mathbf{J_t})}{\|g_S(\mathbf{J_t})\|}(|S_+|\nabla F_\mu(\mathbf{J_t}) - g_{S_-}(\mathbf{J_t}))) \quad (30)$$

$$\geq k\mathbb{E}_\sigma(\frac{1}{n}\sum_{t=0}^T \delta_t \|g_{S_+}(\mathbf{J_t}) - g_{S_-}(\mathbf{J_t})\| - \frac{1}{n}\sum_{t=0}^T \delta_t \||S_+|\nabla F_\mu(\mathbf{J_t}) - g_{S_-}(\mathbf{J_t})\|)$$

$$\geq k\frac{1}{n}\sum_{t=0}^T \delta_t \mathbb{E}_\sigma(\|g_{S_+}(\mathbf{J_t}) - g_{S_-}(\mathbf{J_t})\|) - \sum_{t=0}^T \delta_t \gamma'\|\nabla F_S(\mathbf{J_t})\|$$

$$\geq 0$$

Therefore, combining Equation (27) and (30), we have $\mathbb{E}[\text{gen}^{lin}(\mathbf{J_T})] \leq 2\gamma'\mathbb{V}_m \mathbb{E}R_S(\mathcal{L}_{\mathbf{J}|S})$.

$\square$

**Lemma A.4.** *Given* $\mathbf{J} = \mathcal{A}(S)$, *the formula* $R_S(\mathcal{L}_{\mathbf{J}|S})$ *can be upper bounded with:*

$$R_S(\mathcal{L}_{\mathbf{J}|S}) \leq -\mathbb{E}\int_t \frac{dF_S(\mathbf{J_t})}{\sqrt{n}}\sqrt{1 + \frac{\text{Tr}(\Sigma(\mathbf{w}))}{\|\nabla F_S(\mathbf{w})\|_2^2}}. \quad (31)$$

*Proof.* Let us start with the calculation of $R_S(\mathbf{w}^T\nabla f(\mathbf{J_t}))$:

$$\begin{aligned}
R_S(\{\mathbf{w}^T\nabla f(\mathbf{J_t})|\|\mathbf{w}\| \leq \delta\}) &= \frac{1}{n}\mathbb{E}_\sigma\left(\sup_{\|\mathbf{w}\|\leq\delta} \mathbf{w}^T\sum_{i=1}^n \sigma_i \nabla f(\mathbf{J_t}, z_i)\right) \\
&= \frac{\delta}{n}\mathbb{E}_\sigma\left(\sqrt{\left\|\sum_{i=1}^n \sigma_i \nabla f(\mathbf{J_t}, z_i)\right\|^2}\right) \\
&\leq \frac{\delta}{n}\left(\sqrt{\mathbb{E}_\sigma\left\|\sum_{i=1}^n \sigma_i \nabla f(\mathbf{J_t}, z_i)\right\|^2}\right) \quad (32) \\
&\overset{\blacktriangle}{\leq} \frac{\delta}{n}\left(\sqrt{\mathbb{E}_\sigma\sum_{i=1}^n \|\sigma_i \nabla f(\mathbf{J_t}, z_i)\|^2}\right) \\
&= \frac{\delta}{n}\sqrt{\sum_{i=1}^n \|\nabla f(\mathbf{J_t}, z_i)\|^2},
\end{aligned}$$

where $\blacktriangle$ represents using the relation that for $i, j$ satisfying $i \neq j$, we have $\mathbb{E}\sigma_i\sigma_j = 0$.

Because $w_i$ is independent of $w_j$ if $i \neq j$, we have:

$$R_S(\mathcal{L}_{\mathbf{J}|S}) = R_S(\{f(\mathbf{J_0}) + \sum_{t=0}^{T-1} \mathbf{w_t}^\mathrm{T} \nabla f(\mathbf{J_t}) \mid \|\mathbf{w_t}\| \leq \delta_t\})$$

$$= \sum_{t=0}^{T-1} R_S(\{\mathbf{w_t}^\mathrm{T} \nabla f(\mathbf{J_t}) | \|\mathbf{w_t}\| \leq \delta_t\}) \tag{33}$$

$$\leq \sum_{t=0}^{T-1} \frac{\delta_t}{n} \sqrt{\sum_{i=1}^n \|\nabla f(\mathbf{J_t}, z_i)\|^2}.$$

The covariance of gradient noise can be calculated as:

$$\mathrm{Tr}[\Sigma(\mathbf{w})] = \mathrm{Tr}[\frac{1}{n} \sum_{i=1}^n \nabla f(\mathbf{w}, z_i) \nabla f(\mathbf{w}, z_i)^\mathrm{T} - \nabla F_S(\mathbf{w}) \nabla F_S(\mathbf{w})^\mathrm{T}]$$

$$= \frac{1}{n} \sum_{i=1}^n \mathrm{Tr}[\nabla f(\mathbf{w}, z_i) \nabla f(\mathbf{w}, z_i)^\mathrm{T}] - \mathrm{Tr}[\nabla F_S(\mathbf{w}) \nabla F_S(\mathbf{w})^\mathrm{T}] \tag{34}$$

$$= \frac{1}{n} \sum_{i=1}^n \|\nabla f(\mathbf{w}, z_i)^2\| - \|\nabla F_S(\mathbf{w})\|^2$$

Taking Equation (33) and $\delta_t \triangleq \|\eta_t \nabla F_S(\mathbf{J_t})\|$ into Equation (34), we have :

$$R_S(\mathcal{L}_{\mathbf{J}|S}) \leq \sum_{t=0}^{T-1} \frac{\delta_t}{n} \sqrt{\sum_{i=1}^n \|\nabla f(\mathbf{J_t}, z_i)\|^2}$$

$$= \sum_{t=0}^{T-1} \frac{\delta_t}{\sqrt{n}} \sqrt{\mathrm{Tr}[\Sigma(\mathbf{J_t})] + \|\nabla F_S(\mathbf{J_t})\|^2} \tag{35}$$

$$= \sum_{t=0}^{T-1} \frac{\eta_t \|\nabla F_S(\mathbf{J_t})\|}{\sqrt{n}} \sqrt{\mathrm{Tr}[\Sigma(\mathbf{J_t})] + \|\nabla F_S(\mathbf{J_t})\|^2}$$

When $\eta_t$ is small, $\delta_t \approx -\mathbb{E}_\epsilon \frac{(\mathbf{J_{t+1}} - \mathbf{J_t})^\mathrm{T} \nabla F_S(\mathbf{J_t})}{\|\nabla F_S(\mathbf{J_t})\|} \approx -\mathbb{E}_\epsilon \frac{F_S(\mathbf{J_{t+1}}) - F_S(\mathbf{J_t})}{\|\nabla F_S(\mathbf{J_t})\|}$ holds, therefore we have:

$$\mathbb{E}R_S(\mathcal{L}_{\mathbf{J}|S}) \leq \mathbb{E} \sum_{t=0}^{T-1} \frac{\delta_t}{n} \sqrt{\sum_{i=1}^n \|\nabla f(\mathbf{J_t}, z_i)\|_2^2}$$

$$\approx -\mathbb{E} \sum_{t=0}^{T-1} \frac{F_S(\mathbf{J_{t+1}}) - F_S(\mathbf{J_t})}{\sqrt{n}} \sqrt{1 + \frac{\mathrm{Tr}(\Sigma(\mathbf{w}))}{\|\nabla F_S(\mathbf{w})\|_2^2}} \tag{36}$$

$$\approx -\mathbb{E} \int_t \frac{dF_S(\mathbf{J_t})}{\sqrt{n}} \sqrt{1 + \frac{\mathrm{Tr}(\Sigma(\mathbf{w}))}{\|\nabla F_S(\mathbf{w})\|_2^2}}$$

$\square$

**Theorem A.5.** *Under Assumption 3.4, given $S \sim \mu^n$, let $\mathbf{J} = \mathcal{A}(S)$, where $\mathcal{A}$ denotes the SGD or GD algorithm training with $T$ steps, we have:*

$$\mathbb{E}[F_\mu(\mathbf{J_T}) - F_S(\mathbf{J_T})] \leq -2\gamma' \mathbb{V}_m \mathbb{E} \int_t \frac{dF_S(\mathbf{J_t})}{\sqrt{n}} \sqrt{1 + \frac{\mathrm{Tr}(\Sigma(\mathbf{J_t}))}{\|\nabla F_S(\mathbf{J_t})\|_2^2}} + \mathcal{O}(\eta_m), \tag{37}$$

*where* $\mathbb{V}(\mathbf{w}) = \frac{\|\nabla F_S(\mathbf{w})\|}{\mathbb{E}_{U \subset S}\|\frac{|U|}{n} \nabla F_U(\mathbf{w}) - \frac{|S|-|U|}{n} \nabla F_{S/U}(\mathbf{w})\|}$, $\mathbb{V}_m = \max_t \mathbb{V}(\mathbf{J_t})$ *and* $\gamma' = \max\{1, \max_{U \subset S; t} \frac{|U| \|\nabla F_U(\mathbf{J_t})\|}{n \|\nabla F_S(\mathbf{J_t})\|}\} \gamma$.

*Proof.* We rewrite Equation 2 of the update of SGD with batchsize $b$ here:

$$\mathbf{J_t} = \mathbf{J_{t-1}} - \eta_t \nabla F_S(\mathbf{J_{t-1}}) + \eta_t \epsilon_t \tag{38}$$

where we simplify the $\epsilon(\mathbf{w_t})$ as $\epsilon_t$, then we can expand the function at $f(\mathbf{J_T})$ as:

$$
\begin{aligned}
f^{lin}(\mathbf{J_T}) &\triangleq f(\mathbf{J_0}) + \sum_{t=0}^{T-1}(\eta_t \nabla F_S(\mathbf{J_t}) + \epsilon)^{\mathrm{T}}\nabla f(\mathbf{J_t}) \\
&= f(\mathbf{J_0}) + \sum_{t=0}^{K-1} \eta_t \nabla F_S(\mathbf{J_t})^{\mathrm{T}}\nabla f(\mathbf{J_t}) + \sum_{t=0}^{T-1} \epsilon_t^{\mathrm{T}}\nabla f(\mathbf{J_t})
\end{aligned}
\tag{39}
$$

Note that when the learning rate is small, we have $f(\mathbf{J_T}) \approx f^{lin}(\mathbf{J_T})$.

The difference between the distributional value and the empirical value of the linear function can be calculated as:

$$
\begin{aligned}
&\mathbb{E}[F_\mu(\mathbf{J_0}) + \sum_{t=0}^{T-1}(\eta_t \nabla F_S(\mathbf{J_t}) + \epsilon)^{\mathrm{T}}\nabla F_\mu(\mathbf{J_t})] - \mathbb{E}[F_S(\mathbf{J_0}) + \sum_{t=0}^{T-1}(\eta_t \nabla F_S(\mathbf{J_t}) + \epsilon)^{\mathrm{T}}\nabla F_S(\mathbf{J_t})] \\
&= \mathbb{E}[F_\mu(\mathbf{J_0}) - F_S(\mathbf{J_0}) + \sum_{t=0}^{T-1}(\eta_t \nabla F_S(\mathbf{J_t}) + \epsilon)^{\mathrm{T}}\nabla F_\mu(\mathbf{J_t}) - \sum_{t=0}^{T-1}(\eta_t \nabla F_S(\mathbf{J_t}) + \epsilon)^{\mathrm{T}}\nabla F_S(\mathbf{J_t})] \\
&= \mathbb{E}[\sum_{t=0}^{T-1}\eta_t \nabla F_S(\mathbf{J_t})^{\mathrm{T}}(\nabla F_\mu(\mathbf{J_t}) - \nabla F_S(\mathbf{J_t})) + \sum_{t=0}^{T-1}\epsilon_t^{\mathrm{T}}(\nabla F_\mu(\mathbf{J_t}) - \nabla F_S(\mathbf{J_t}))]] \\
&\stackrel{\blacktriangle}{=} \mathbb{E}[\sum_{t=0}^{T-1}\eta_t \nabla F_S(\mathbf{J_t})^{\mathrm{T}}(\nabla F_\mu(\mathbf{J_t}) - \nabla F_S(\mathbf{J_t}))] \\
&\leq \Phi(S, \mathcal{F}_{\mathbf{J}|S}),
\end{aligned}
\tag{40}
$$

where $\blacktriangle$ using the equation that $\mathbb{E}[\epsilon_t^{\mathrm{T}}(\nabla F_\mu(\mathbf{J_t}) - \nabla F_S(\mathbf{J_t}))] = 0$, according to Equation 20.

Because of $\mathbb{E}[F_\mu(\mathbf{J_T}) - F_S(\mathbf{J_T})] = \mathbb{E}[F_\mu^{lin}(\mathbf{J_T}) + \mathcal{O}(\eta_m) - F_S^{lin}(\mathbf{J_T}) - \mathcal{O}(\eta_m)] = \mathbb{E}[F_\mu^{lin}(\mathbf{J_T}) - F_S^{lin}(\mathbf{J_T})] + \mathcal{O}(\eta_m)$(from Proposition A.1), by applying Lemma A.3 and Lemma A.4, the theorm is proved.

$\square$

**Corollary A.6.** *If function $f(\cdot)$ is $\beta$-smooth, under Assumption 3.4 given $S \sim \mu^n$, let $\mathbf{J} = \mathcal{A}(S)$, $\eta_t = \frac{c}{\beta(t+1)}$, $M_2^2 = \max_t \mathbb{E}_{t-1}(\|\nabla F_S(\mathbf{J_t}) + \epsilon(\mathbf{J_t})\|^2)$ and $M_4^4 = \max_t \mathbb{E}_{t-1}(\|\nabla F_S(\mathbf{J_t}) + \epsilon(\mathbf{J_t})\|^4)$ , where $\mathcal{A}$ denoted the SGD or GD algorithm training with $T$ steps, we have:*

$$
\begin{aligned}
\mathbb{E}[F_\mu(\mathbf{J_T}) - F_S(\mathbf{J_T})] \leq &- 2\gamma'\mathbb{V}_m\mathbb{E}\int_t \frac{dF_S(\mathbf{J_t})}{\sqrt{n}}\sqrt{1 + \frac{\mathrm{Tr}(\Sigma(\mathbf{J_t}))}{\|\nabla F_S(\mathbf{J_t})\|_2^2}} \\
&+ 2c^2\gamma'\mathbb{V}_m M_4^2\sqrt{\mathbb{E}\int_t \frac{dt}{n\beta^2(t+1)^4}\left(1 + \frac{\mathrm{Tr}(\Sigma(\mathbf{J_t}))}{\|\nabla F_S(\mathbf{J_t})\|_2^2}\right)} \\
&+ 2c^2\frac{M_2^2}{\beta}.
\end{aligned}
\tag{41}
$$

*where* $\mathbb{V}(\mathbf{w}) = \frac{\|\nabla F_S(\mathbf{w})\|}{\mathbb{E}_{U \subset S}\|\frac{|U|}{n}\nabla F_U(\mathbf{w}) - \frac{|S|-|U|}{n}\nabla F_{S/U}(\mathbf{w})\|}$, $\mathbb{V}_m = \max_t \mathbb{V}(\mathbf{J_t})$ *and* $\gamma' = \max\{1, \max_{U \subset S; t} \frac{|U|\|\nabla F_U(\mathbf{J_t})\|}{n\|\nabla F_S(\mathbf{J_t})\|}\}\gamma$.

*Proof.* If $f(\cdot)$ is $\beta$-smooth, we have:

$$f(\mathbf{J_{t+1}}) - f(\mathbf{J_t}) \leq (\mathbf{J_{t+1}} - \mathbf{J_t})^{\mathrm{T}}\nabla f(\mathbf{J_t}) + \frac{1}{2}\beta\|\mathbf{J_{t+1}} - \mathbf{J_t}\|^2 \tag{42}$$

$$f(\mathbf{J_{t+1}}) - f(\mathbf{J_t}) \geq (\mathbf{J_{t+1}} - \mathbf{J_t})^{\mathrm{T}}\nabla f(\mathbf{J_t}) - \frac{1}{2}\beta\|\mathbf{J_{t+1}} - \mathbf{J_t}\|^2. \tag{43}$$

Combining the two equations, we obtain:

$$|R_\mu(\mathbf{J_T}) - R_S(\mathbf{J_T})| \leq |F_\mu(\mathbf{J_T}) - F_\mu^{lin}(\mathbf{J_T})| + |F_S(\mathbf{J_T}) - F_S^{lin}(\mathbf{J_T})|$$

$$\leq \frac{\beta}{2}\sum_{t=0}^{T-1}\|\mathbf{J_{t+1}} - \mathbf{J_t}\|^2 + \frac{\beta}{2}\sum_{t=0}^{T-1}\|\mathbf{J_{t+1}} - \mathbf{J_t}\|^2$$

$$= \beta\sum_{t=0}^{T-1}\|\mathbf{J_{t+1}} - \mathbf{J_t}\|^2. \tag{44}$$

The generalization error can be divided into three parts:

$$\mathbb{E}[F_\mu(\mathbf{J_T}) - F_S(\mathbf{J_T})] \leq -2\gamma'\mathbb{V}_m\mathbb{E}\int_t \frac{dF_S(\mathbf{J_t})}{\sqrt{n}}\sqrt{1 + \frac{\text{Tr}(\Sigma(\mathbf{J_t}))}{\|\nabla F_S(\mathbf{J_t})\|_2^2}}$$

$$\underbrace{-2\gamma'\mathbb{V}_m\mathbb{E}\int_t \frac{dF_S^{lin}(\mathbf{J_t}) - dF_S(\mathbf{J_t})}{\sqrt{n}}\sqrt{1 + \frac{\text{Tr}(\Sigma(\mathbf{J_t}))}{\|\nabla F_S(\mathbf{J_t})\|_2^2}}}_{(A)} \tag{45}$$

$$\underbrace{+ \beta\mathbb{E}\sum_{t=0}^{T-1}\|\mathbf{J_{t+1}} - \mathbf{J_t}\|^2}_{(B)}.$$

The term "$(A)$" is caused by using $F_S(\mathbf{J_{t+1}}) - F_S(\mathbf{J_t})$ to replace $F_S^{lin}(\mathbf{J_{t+1}}) - F_S^{lin}(\mathbf{J_t})$. The term "$(B)$" is induced by $\text{gen}^{nl}(\mathbf{J_T})$. Then, we want to give a upper bound of $(A)$ using $M_4^4$:

$$(A) \overset{(\star)}{\leq} 2\gamma'\mathbb{V}_m\mathbb{E}\sum_{t=0}^{T-1}\frac{\beta\|\mathbf{J_{t+1}} - \mathbf{J_t}\|^2}{\sqrt{n}}\sqrt{1 + \frac{\text{Tr}(\Sigma(\mathbf{J_t}))}{\|\nabla F_S(\mathbf{J_t})\|_2^2}}$$

$$\overset{(\star\star)}{\leq} 2c^2\gamma'\mathbb{V}_m\mathbb{E}\sum_{t=0}^{T-1}\frac{\|\nabla F_S(\mathbf{J_t}) + \epsilon(\mathbf{J_t})\|^2}{\beta\sqrt{n}(t+1)^2}\sqrt{1 + \frac{\text{Tr}(\Sigma(\mathbf{J_t}))}{\|\nabla F_S(\mathbf{J_t})\|_2^2}}$$

$$= 2c^2\gamma'\mathbb{V}_m\sum_{t=0}^{T-1}\mathbb{E}_{t-1}\frac{\|\nabla F_S(\mathbf{J_t}) + \epsilon(\mathbf{J_t})\|^2}{\beta\sqrt{n}(t+1)^2}\sqrt{1 + \frac{\text{Tr}(\Sigma(\mathbf{J_t}))}{\|\nabla F_S(\mathbf{J_t})\|_2^2}}$$

$$\overset{(\star\star\star)}{\leq} 2c^2\gamma'\mathbb{V}_m\sum_{t=0}^{T-1}\sqrt{\frac{\mathbb{E}_{t-1}\|\nabla F_S(\mathbf{J_t}) + \epsilon(\mathbf{J_t})\|^4}{\beta^2 n(t+1)^4}\mathbb{E}_{t-1}\left(1 + \frac{\text{Tr}(\Sigma(\mathbf{J_t}))}{\|\nabla F_S(\mathbf{J_t})\|_2^2}\right)} \tag{46}$$

$$\leq 2c^2\gamma'\mathbb{V}_m\sum_{t=0}^{T-1}\sqrt{\frac{M_4^4}{\beta^2 n(t+1)^4}\mathbb{E}_{t-1}\left(1 + \frac{\text{Tr}(\Sigma(\mathbf{J_t}))}{\|\nabla F_S(\mathbf{J_t})\|_2^2}\right)}$$

$$\leq 2c^2\gamma'\mathbb{V}_m M_4^2\sqrt{\mathbb{E}\sum_{t=0}^{T-1}\frac{1}{\beta^2 n(t+1)^4}\left(1 + \frac{\text{Tr}(\Sigma(\mathbf{J_t}))}{\|\nabla F_S(\mathbf{J_t})\|_2^2}\right)}.$$

where ($\star$) is due to the Equation 44, ($\star\star$) is due to the update rule of $\mathbf{J_t}$ and ($\star\star\star$) is que to Hölder's inequality. In the following, we use $M_2^2$ to give a upper bound for $(B)$:

$$
\begin{aligned}
(B) &\leq \frac{c^2}{\beta} \sum_{t=0}^{T-1} \frac{1}{(t+1)^2} \mathbb{E}\|\nabla F(\mathbf{J_t}) + \epsilon(\mathbf{J_t})\|^2 \\
&\leq \frac{c^2}{\beta} \sum_{t=0}^{T-1} \frac{1}{(t+1)^2} M_2^2 \\
&\leq \frac{c^2}{\beta} \left( M_2^2 + \sum_{t=1}^{T-1} \frac{1}{(t+1)^2} M_2^2 \right) \\
&\leq \frac{c^2}{\beta} \left( M_2^2 + \sum_{t=1}^{T-1} \left( \frac{1}{t} - \frac{1}{t+1} \right) M_2^2 \right) \\
&\leq \frac{c^2}{\beta} \left( 2M_2^2 - \frac{1}{T} M_2^2 \right) \\
&\leq 2c^2 \frac{M_2^2}{\beta}
\end{aligned}
\tag{47}
$$

Taking the upper bound value of "(A)" and "(B)" into Equation 45, we obtain the result.

$\square$

## B  Relaxed Assumption and Corresponding Bound

**Assumption B.1.** There is a value $\gamma$, $T_0$ and $\zeta$, so that for all $\mathbf{w} \in \{\mathbf{J_t}|t \in \mathbb{N} \wedge t < T_0\}$, we have $\|\nabla F_\mu(\mathbf{w})\| \leq \gamma\|\nabla F_S(\mathbf{w})\|$ and for all $\mathbf{w} \in \{\mathbf{J_t}|t \in \mathbb{N} \wedge t \geq T_0\}$, we have $\|\nabla F_\mu(\mathbf{w})\| \leq \gamma\|\nabla F_S(\mathbf{w})\| + \zeta$.

**Theorem B.2.** *Under Assumption B.1, given $S \sim \mu^n$, let $\mathbf{J} = \mathcal{A}(S)$, where $\mathcal{A}$ denotes the SGD or GD algorithm training with $T$ steps, we have:*

$$
\mathbb{E}[F_\mu(\mathbf{J_T}) - F_S(\mathbf{J_T})] \leq -2\gamma' \mathbb{V}_m \mathbb{E} \int_t \frac{dF_S(\mathbf{J_t})}{\sqrt{n}} \sqrt{1 + \frac{\mathrm{Tr}(\Sigma(\mathbf{J_t}))}{\|\nabla F_S(\mathbf{J_t})\|_2^2}} + \frac{1}{2} \sum_{t=T_0}^{T} \eta_t \|\nabla F_S(\mathbf{J_t})\| \zeta + \mathcal{O}(\eta_m),
$$

$$\tag{48}$$

*where* $\mathbb{V}(\mathbf{w}) = \dfrac{\|\nabla F_S(\mathbf{w})\|}{\mathbb{E}_{U \subset S} \|\frac{|U|}{n} \nabla F_U(\mathbf{w}) - \frac{n-|U|}{n} \nabla F_{S/U}(\mathbf{w})\|}$, $\mathbb{V}_m = \max_t \mathbb{V}(\mathbf{J_t})$ *and* $\gamma' = \max\{1, \max\limits_{U \subset S; t} \dfrac{|U| \|\nabla F_U(\mathbf{J_t})\|}{n \|\nabla F_S(\mathbf{J_t})\|}\} \gamma$.

*Proof.* Most of the proofs in this part are the same as those in Appendix A, except for Equation 30. The Equation 30 is replaced by:

$$k\mathbb{E}_\sigma \sup_{h\in\mathcal{L}_{\mathbf{J}|S}} (\frac{1}{n}\sum_i^n \sigma_i h(z_i)) - \mathbb{E}_{S',\sigma} \sup_{h\in\mathcal{F}_{\mathbf{J}|S}} (\frac{1}{n}\sum_i^n \sigma_i(h(z_i)))$$

$$= k\mathbb{E}_\sigma(\frac{1}{n}\sum_{t=0}^{\mathrm{T}}\delta_t\|g_{S_+}(\mathbf{J_t}) - g_{S_-}(\mathbf{J_t})\|) - \mathbb{E}_{S',\sigma}(\frac{1}{n}\sum_{t=0}^{T}\delta_t\frac{g_S(\mathbf{J_t})}{\|g_S(\mathbf{J_t})\|}(g_{S'_+}(\mathbf{J_t}) - g_{S_-}(\mathbf{J_t})))$$

$$= k\mathbb{E}_\sigma(\frac{1}{n}\sum_{t=0}^{\mathrm{T}}\delta_t\|g_{S_+}(\mathbf{J_t}) - g_{S_-}(\mathbf{J_t})\| - \frac{1}{n}\sum_{t=0}^{T}\delta_t\frac{g_S(\mathbf{J_t})}{\|g_S(\mathbf{J_t})\|}(|S_+|\nabla F_\mu(\mathbf{J_t}) - g_{S_-}(\mathbf{J_t})))$$

$$\geq k\mathbb{E}_\sigma(\frac{1}{n}\sum_{t=0}^{\mathrm{T}}\delta_t\|g_{S_+}(\mathbf{J_t}) - g_{S_-}(\mathbf{J_t})\| - \frac{1}{n}\sum_{t=0}^{T}\delta_t\||S_+|\nabla F_\mu(\mathbf{J_t}) - g_{S_-}(\mathbf{J_t})\|) \qquad (49)$$

$$\geq k\frac{1}{n}\sum_{t=0}^{\mathrm{T}}\delta_t\mathbb{E}_\sigma(\|g_{S_+}(\mathbf{J_t}) - g_{S_-}(\mathbf{J_t})\|) - \sum_{t=0}^{T}\delta_t\gamma'\|\nabla F_S(\mathbf{J_t})\| - \frac{1}{2}\sum_{t=T_0}^{T}\delta_t\zeta$$

$$\geq -\frac{1}{2}\sum_{t=T_0}^{T}\eta_t\|\nabla F_S(\mathbf{J_t})\|\zeta.$$

$\square$

*Remark* B.3. Compared of Theorem 3.6, we have a extra term $\sum_{t=T_0}^{T}\eta_t\|\nabla F_S(\mathbf{J_t})\|\zeta$ here. Since the unrelaxed assumption $\|\nabla F_\mu(\mathbf{w})\| \leq \gamma\|\nabla F_S(\mathbf{w})\|$ is not satisfied only when $\|\nabla F_S(\mathbf{w})\|$ is relative small, the term $\sum_{t=T_0}^{T}\eta_t\|\nabla F_S(\mathbf{J_t})\|\zeta$ is small value.

## C  Experiments

### C.1  Calculation of $\mathcal{C}(\mathbf{J})$

To reduce the calculation, we construct a randomly sampled subset $S_{\mathrm{sp}} = \{z_1^{\mathrm{sp}}, ..., z_n^{\mathrm{sp}}\} \subset S$.

$$\int_t \frac{dF_S(\mathbf{J_t})}{\sqrt{n}}\sqrt{1 + \frac{\mathrm{Tr}(\Sigma(\mathbf{J_t}))}{\|\nabla F_S(\mathbf{J_t})\|_2^2}} = \int_t \frac{dF_S(\mathbf{J_t})}{\sqrt{n}}\sqrt{\frac{\sum_{i=1}^{n}\|\nabla f(\mathbf{J_t}, z_i)\|_2^2}{n}\frac{1}{\|\nabla F_S(\mathbf{J_t})\|_2^2}}$$

$$\approx \int_t \frac{dF_S(\mathbf{J_t})}{\sqrt{n}}\sqrt{\frac{\sum_{i=1}^{n_{\mathrm{sp}}}\|\nabla f(\mathbf{J_t}, z_i^{\mathrm{sp}})\|_2^2}{n_{\mathrm{sp}}}\frac{1}{\|\nabla F_S(\mathbf{J_t})\|_2^2}}$$

Denote the weights after $t$-epoch training as $\mathbf{X_t}$. We can roughly calculated $\mathcal{C}(\mathbf{J_t})$

$$\sum_{t=1}^{T}\frac{F_S(\mathbf{X_t}) - F_S(\mathbf{X_{t-1}})}{\sqrt{n}}\sqrt{\frac{\sum_{i=1}^{n_{\mathrm{sp}}}\|\nabla f(\mathbf{X_t}, z_i^{\mathrm{sp}})\|_2^2}{n_{\mathrm{sp}}}\frac{1}{\|\nabla F_S(\mathbf{X_t})\|_2^2}}$$

### C.2  Experimental Details

Here, we give a detail setting of the experiment for each figure.

Figure 1   The learning rate is fixed to 0.05 during all the training process. The batch size is 256. All experiments is trained with 100 epoch. The test accuracy for CIFAR-10, CIFAR-100, and SVHN are 87.64%, 55.08%, and 92.80%, respectively.

Figure 2   The initial learning rate is set to 0.05 with the batch size of 1024. We use the Cosine Annealing LR Schedule to adjust the learning rate during training.

Figure 3   Each point is an average of three repeated experiments. We stop training when the training loss is small than 0.2.

## C.3 Experimental exploration of $\text{gen}^{nl}(\mathbf{J_T})$

In this section, our aim is to investigate the conditions under which $\text{gen}^{nl}(\mathbf{J_T}) \approx 0$. Since directly calculating the difference $|R_\mu(\mathbf{J_T}) - R_S(\mathbf{J_T})|$ is challenging, we concentrate on the upper bound value $|R_\mu(\mathbf{J_T})| + |R_S(\mathbf{J_T})|$.

We conduct the experiment using cifar10-5k dataset and fc-tanh network, following the setting of paper [9]. Cifar10-5k[9] is a subset of cifar10 dataset. Building upon the work of [1], we compute the Relative Progress Ratio (RP) and Test Relative Progress Ratio (TRP) throughout the training process. We initially consider the case of gradient descent. The definitions of RP and TRP for gradient descent are as follows:

$$\text{RP}(\mathbf{J_t}) \triangleq \frac{F_S(\mathbf{J_{t+1}}) - F_S(\mathbf{J_t})}{\eta \|\nabla F_S(\mathbf{J_t})\|^2} \tag{50}$$

$$\text{TRP}(\mathbf{J_t}) \triangleq \frac{F_{S'}(\mathbf{J_{t+1}}) - F_{S'}(\mathbf{J_t})}{\eta \nabla F_S(\mathbf{J_t})^{\text{T}} \nabla F_{S'}(\mathbf{J_t})}. \tag{51}$$

Therefore, we have:

$$
\begin{aligned}
&F_S(\mathbf{J_0}) + \sum_{t=1}^{T}(F_S(\mathbf{J_t}) - F_S(\mathbf{J_{t-1}})) - F_S(\mathbf{J_0}) - \sum_{t=0}^{T-1}(\mathbf{J_t} - \mathbf{J_{t-1}})^{\text{T}}\nabla F_S(\mathbf{J_{t-1}}) \\
&= \sum_{t=1}^{T}\left[(F_S(\mathbf{J_t}) - F_S(\mathbf{J_{t-1}})) - (\mathbf{J_t} - \mathbf{J_{t-1}})^{\text{T}}\nabla F_S(\mathbf{J_{t-1}})\right] \\
&= \sum_{t=1}^{T}\left[(F_S(\mathbf{J_t}) - F_S(\mathbf{J_{t-1}})) + \eta\|\nabla F_S(\mathbf{J_{t-1}})\|^2\right] \\
&= \sum_{t=1}^{T}\left[\eta_t(1 + \text{RP}(\mathbf{J_{t-1}}))\|\nabla F_S(\mathbf{J_{t-1}})\|^2\right]
\end{aligned}
\tag{52}
$$

Following the same way, we have:

$$
\begin{aligned}
&F_{S'}(\mathbf{J_0}) + \sum_{t=1}^{T}(F_{S'}(\mathbf{J_t}) - F_{S'}(\mathbf{J_{t-1}})) - F_{S'}(\mathbf{J_0}) - \sum_{t=0}^{T-1}(\mathbf{J_t} - \mathbf{J_{t-1}})^{\text{T}}\nabla F_{S'}(\mathbf{J_{t-1}}) \\
&= \sum_{t=1}^{T}\left[\eta_t(1 + \text{TRP}(\mathbf{J_{t-1}}))\nabla F_S(\mathbf{J_{t-1}})^{\text{T}}\nabla F_{S'}(\mathbf{J_{t-1}})\right]
\end{aligned}
\tag{53}
$$

Combining Equation (52) and Equation (53), we have:

$$
\begin{aligned}
&|\text{gen}^{nl}(\mathbf{J_T})| \leq \\
&\sum_{t=1}^{T}\eta_t\left[(1 + \text{TRP}(\mathbf{J_{t-1}}))|\nabla F_S(\mathbf{J_{t-1}})^{\text{T}}\nabla F_{S'}(\mathbf{J_{t-1}})| + (1 + \text{RP}(\mathbf{J_{t-1}}))\|\nabla F_S(\mathbf{J_{t-1}})\|^2\right]
\end{aligned}
\tag{54}
$$

Therefore, if we have for all $t$, $\text{RP}(\mathbf{J_t}) \approx -1$ and $\text{TRP}(\mathbf{J_t}) \approx -1$, then $|\text{gen}^{nl}(\mathbf{J_T})| \approx 0$.

From Figure 4 we find that in stable regime, where the sharpness is below the $\frac{2}{\eta}$, we have $\text{TRP} \approx \text{RP} \approx -1$. Under small learning rate, the gradient descent doesn't enter the regime of edge of stability and we have $\text{TRP} \approx \text{RP} \approx -1$ during whole training process and $\text{gen}^{nl}(\mathbf{J_T}) \approx 0$.

Next, we consider the case of Stochastic Gradient Descent (SGD). Due to the stochastic estimation of the gradient, we need to rely on some approximations. Let $\mathbf{X_t^i}$ represent the weights after the $t$-epoch and $i$-th iteration of training. We assume a constant learning rate $\eta$ for SGD. The gradient is approximated as follows:

$$\eta\nabla F_S(\mathbf{X_t^i}) \approx \frac{B}{n}(\mathbf{X_t} - \mathbf{X_{t+1}}) = \frac{B}{n}\sum_{i=1}^{\frac{n}{B}}\nabla F_S(\mathbf{X_t^i}), \tag{55}$$

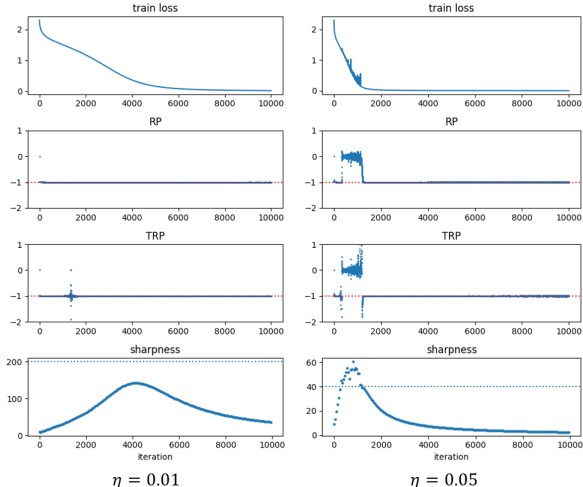

Figure 4: **Exploration of** $\text{gen}^{nl}(\mathbf{J_T})$ **on Gradient descent**. Experiments is conducted on cifar10-5k dataset with cross entropy loss. The blue dash line in fourth row denotes $\frac{2}{\eta}$. Gradient descent enter the EoS regime when the sharpness is above $\frac{2}{\eta}$. Both RP and TRP have values around -1 when sharpness is below the $\frac{2}{\eta}$.

and we appximate $\nabla F_{S'}(\mathbf{X_t^i})$ as:

$$\eta \nabla F_S(\mathbf{X_t^i}) \approx \eta \nabla F_S(\mathbf{X_t}). \tag{56}$$

Therefore, we have:

$$\text{RP}(\mathbf{X_t}) \approx \frac{\eta(F_S(\mathbf{X_{t+1}}) - F_S(\mathbf{X_t}))}{\|\mathbf{X_{t+1}} - \mathbf{X_t}\|} \tag{57}$$

$$\text{TRP}(\mathbf{X_t}) \approx \frac{F_{S'}(\mathbf{X_{t+1}}) - F_{S'}(\mathbf{X_t})}{(\mathbf{X_t} - \mathbf{X_{t+1}})^{\mathrm{T}} \nabla F_{S'}(\mathbf{X_t})}. \tag{58}$$

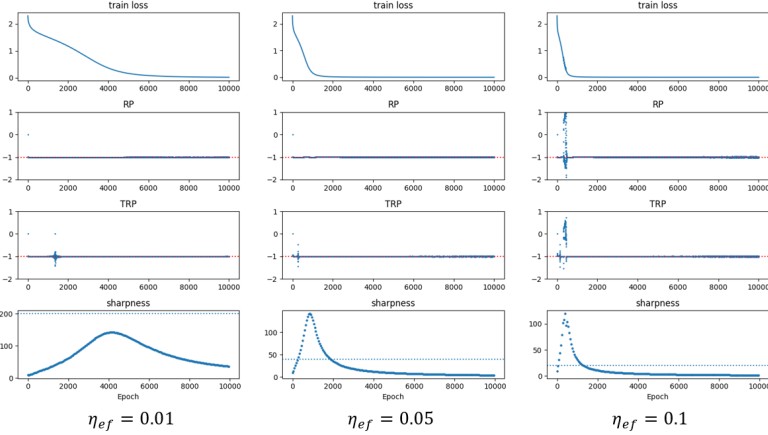

Figure 5: **Exploration of** $\text{gen}^{nl}(\mathbf{J_T})$ **on SGD case.** Here, the effective learning rate is defined as $\eta_{ef} \triangleq \frac{n}{B}\eta$. We still have $\text{gen}^{nl}(\mathbf{J_T}) \approx 0$ under small learning rate.

We calculated the effect learning rate for SGD as $\eta_{ef} \triangleq \frac{n}{B}\eta$. Figure 5 shows that the conclusions of SGD are similar as GD, except that the conditions of entering EoS are different.

Table 4: **Comparison of trajectory based generalization bounds.** Only our proposed method can apply to the SGD with rich trajectory related information.

| Method | Conditions | T.R.T |
|---|---|---|
| Nikolakakis et al. [25] | Gradient Descent, $\eta_t \leq \frac{c}{t} \leq \frac{1}{\beta}$, $\beta$-smooth | $\sum_{t=1}^{T} \eta_t \frac{1}{n} \sum_{i=1}^{n} \|\nabla f(\mathbf{J_t}, z_i)\|^2$ |
| Neu et al. [24] | $\beta$-smooth, $\mathbb{E}\left[\|\nabla f(\mathbf{w}, z) - \nabla F_\mu(\mathbf{w})\|\right] \leq v$, $f(\cdot)$ is subguassian distribution | $\sqrt{T\eta^2}$ |
| Park et al. [27] | Weak Lipschitz continuity, Piecewise $\beta'$-smooth, $f(\cdot)$ is bounded, $\eta < \frac{2}{\beta}$ | $T$ |
| Ours | Small Learning Rate, $\|\nabla F_\mu(\mathbf{w})\| \leq \gamma \|\nabla F_S(\mathbf{w})\|$ | $\int_t dF_S(\mathbf{J_t}) \sqrt{1 + \frac{\text{Tr}(\Sigma(\mathbf{J_t}))}{\|\nabla F_S(\mathbf{J_t})\|^2}}$ |

# D  Other Related Work

This part compares the works that is not listed in Table 2. Table 4 gives other trajectory based generalization bounds. [25] is a stability based work designed mainly for generalization of gradient descent. It removes the Lipschitz assumption, and replaced by the term $\sum_{t=1}^{T} \eta_t \frac{1}{n} \sum_{i=1}^{n} \|\nabla f(\mathbf{J_t}, z_i)\|^2$ in the generalization bounds. This helps enrich the trajectory information in the bounds. The limitation of this work is that it can only apply to the gradient descent and it is hard to extend to the stochastic gradient descent. Neu et al. [24] adapt the information-theretical generalization bound to the stochastic gradient descent. The Theorem 1 in Neu et al. [24] contains rich information about the learning trajectory, but most is about $\nabla F_\mu(\mathbf{w})$, which is unavailable for us. Therefore, we mainly consider the result of Corollary 2 in Neu et al. [24], which removes the term $\nabla F_\mu(\mathbf{w})$ by the assumption listed in Table 4. For this Collorary, the remained information within trajectory is merely the $\sqrt{T\eta^2}$. Althouht Neu et al. [24] dosen't require the assumption of small learning rate, the bound contains the dimension of model, which is large for deep neural network. Compared with these work, our proposed method has advantage in that it can both reveal rich information about learning trajectory and applied to stochastic gradient descent.

Chandramoorthy et al. [8] analyzes the generalization behavior based on statistical algorithmic stability. The proposed generalization bound can be applied into algorithms that don't converge. Let $S^{(i)}$ be the dataset obtained by replace $z_i$ in $S$ with another sample $z_i'$ draw from distribution $\mu$. The generalization bound relies on the stability measure $m \triangleq \sup\{\frac{1}{T}\sum_{t=0}^{T-1} f(\mathbf{J_t}|S, z) - \frac{1}{T}\sum_{t=0}^{T-1} f(\mathbf{J_t}|S^{(i)}, z)|z \in \mathcal{Z}, i \in [n]\}$. We don't directly compare with this method because the calculation of $m$ relies on $S^{(i)}$ which contains sample outside of $S$. Therefore, we treat this result as intermediate results. More assumption is needed to remove this dependence of the information about the unseen samples, *i.e.*, the samples outside set $S$.

# E  Effect of Learning Rate and Stochastic Noise

In this part, we want to analyze how learning rate and the stochastic noise jointly affect our proposed generalization bound. Specifically, we denote $p_t(\mathbf{w})$ as the distribution of the $\mathbf{J_t}$ during the training with multiple training steps. Following the work [12], we consider the SDE function as an approximation, which is shown as below:

$$d\mathbf{w} = -\nabla F_S(\mathbf{w})dt + \sqrt{\eta}C^{\frac{1}{2}}d\mathbf{W}(t). \tag{59}$$

The SDE can be regarded as the continuous counterpart of Equation(3) when sets the distribution of noise term $\epsilon'$ in Equation(3) as Gaussian distribution. The influence of the noise $\epsilon$ on $p_t(\mathbf{w})$ is shown in the following theorem.

**Theorem E.1.** *When the updating of the weight $w$ follows Equation (59), the covariance matrix $C$ is a hessian matrix of a function with a scalar output, then we have:*

$$\frac{\partial p_t(\mathbf{w})}{\partial t} = -\sum_{i=1}^{d} \frac{\partial}{\partial \mathbf{w_i}}[\nabla F_S(\mathbf{w})p_t(\mathbf{w}) - \frac{\eta}{2}[\nabla \text{Tr}(C(\mathbf{w})) + \underbrace{C(\mathbf{w})\nabla_w \log(p_t(\mathbf{w}))}_{dampling\ factor}]p_t(\mathbf{w})]. \tag{60}$$

*Remark* E.2. Previous studies [45, 32, 12] tell that the covariance matrix $C$ is proximately equal to the hessian matrix of the loss function with respect to the parameters of DNN. Thus, the above condition that the covariance matrix $C$ is a hessian matrix of a function with scalar output is easy to be satisfied. Formula (60) contains three parts. The item $F_S(\mathbf{w})p_t(\mathbf{w})$ enlarge the probability of parameters being located in the parameter space with low $F_S(\mathbf{w})$. $\nabla\text{Tr}(C(\mathbf{w}))$ and $C(\mathbf{w})\nabla_w \log(p_t(\mathbf{w}))$ ususally

contradict with each other. $\nabla \text{Tr}(C(\mathbf{w}))$ enlarge the probability of parameters being located in the parameter space with low $\text{Tr}(C(\mathbf{w}))$ value, while $C(\mathbf{w})\nabla_w \log(p_t(\mathbf{w}))$ serves as a damping factor to prevent the probability from concentrating on a small space. Therefore, setting larger learning rate gives stronger force for the weight to the area with lower $\text{Tr}(C(\mathbf{w}))$ values. According to Equation 5, we also have a lower $\Sigma(\mathbf{w})$. As a result, large learning rate causes a small lower bound in Theorem 3.6

*Proof.* Based on the condition described above, we can infer that $C(\mathbf{w}) = \nabla\nabla G(\mathbf{w})$, where G is a function with a scalar output.

We first prove that $\nabla \cdot C(\mathbf{w}) = \nabla \text{Tr}(C(\mathbf{w}))$ as below:

$$
\begin{aligned}
[\nabla \cdot C(\mathbf{w})]_j &= [\nabla \cdot \nabla\nabla G(\mathbf{w})]_j \\
&= \sum_i \frac{\partial}{\partial w_i} \frac{\partial}{\partial w_i} \frac{\partial}{\partial w_j} G(\mathbf{w}) \\
&= \frac{\partial}{\partial w_j} \sum_i \frac{\partial}{\partial w_i} \frac{\partial}{\partial w_i} G(\mathbf{w}) \\
&= \frac{\partial}{\partial w_j} \text{Tr}(C(\mathbf{w})).
\end{aligned}
\tag{61}
$$

So far, we can infer that $\nabla \cdot C = \nabla \text{Tr}(C)$. According to Fokker-Planck equation( [26]), we have:

$$
\begin{aligned}
\frac{\partial p_t(\mathbf{w})}{\partial t} &= -\sum_{i=1}^d \frac{\partial}{\partial \mathbf{w_i}} [\nabla F_D(\mathbf{w})p_t(\mathbf{w})] + \frac{1}{2}\eta \sum_{i=1}^d \frac{\partial}{\partial \mathbf{w_i}} \left[ \sum_j \frac{\partial}{\partial \mathbf{w_j}} [C(\mathbf{w})p_t(\mathbf{w})] \right] \\
&= -\sum_{i=1}^d \frac{\partial}{\partial \mathbf{w_i}} [\nabla F_D(\mathbf{w})p_t(\mathbf{w})] + \frac{1}{2}\eta \sum_{i=1}^d \frac{\partial}{\partial \mathbf{w_i}} [p_t(\mathbf{w})\nabla \cdot C + p_t(\mathbf{w})C\nabla_w \log p_t(\mathbf{w})] \\
&= -\sum_{i=1}^d \frac{\partial}{\partial \mathbf{w_i}} \left[ \nabla F_D(\mathbf{w})p_t(\mathbf{w}) - \frac{1}{2}\eta \left[ \nabla \cdot C(\mathbf{w}) + C(\mathbf{w})\nabla_w \log p_t(\mathbf{w}) \right] p_t(\mathbf{w}) \right] \\
&= -\sum_{i=1}^d \frac{\partial}{\partial \mathbf{w_i}} \left[ \nabla F_D(\mathbf{w})p_t(\mathbf{w}) - \frac{1}{2}\eta \left[ \nabla \text{Tr}(C(\mathbf{w})) + C(\mathbf{w})\nabla_w \log p_t(\mathbf{w}) \right] p_t(\mathbf{w}) \right].
\end{aligned}
\tag{62}
$$

Therefore, the theorem is proven. $\square$

