# OpenReview forum: "Learning Trajectories are Generalization Indicators"
_NeurIPS.cc/2023/Conference — NeurIPS 2023 poster_

### Official Review · Reviewer_pZNy · 2023-06-30

**Soundness:** 4 excellent
**Presentation:** 3 good
**Contribution:** 4 excellent
**Rating:** 7
**Confidence:** 3

**Summary:**

The paper introduces a novel generalization bound that incorporates trajectory information, aimed at providing deeper insights than existing methods on generalization at different points during training. The key idea is to analyze the increase in generalization error at each point in training by linearizing the network. Experimental results confirm the effectiveness of the proposed approach in capturing generalization error throughout the training process, even with adjustments to learning rates and label noise levels.

**Strengths:**

**Originality**
As far as I am aware, this is the first generalization bound of its kind; I'm not aware of any other that uses the approach used in this paper. A major strength of the paper is that the derivation of the bound and bound itself are simple to understand, but still seem relatively powerful.

**Quality**
The theory in this paper appears solid. Experiments are also conducted well; testing on VGG13 trained on CIFAR-10 is a good choice. The authors experimentally validate the theoretical assumptions, which also is a major plus. I also view the fact that the bounds predict empirical generalization performance in Figure 3 as a major strength.

**Clarity**
Generally speaking, the paper is understandable.

**Significance**
The paper seems to take a relatively different approach than other works deriving generalization bounds for neural networks. Thus, I view the main significance of the paper as providing a new set of theoretical techniques. This paper is likely to be of significance to the field of developing practical generalization bounds for neural networks.

**Weaknesses:**

In my mind, the main drawback of the paper is that it's hard to establish the significance of the proven results without comparing the numerical generalization bounds implied by Theorem 3.6 with those proved by prior work. Table 2 is helpful, but it would also be good to establish the relative tightness of these bounds numerically if possible. Also, as the authors point out, the small learning rate assumption is restrictive, but also in line with some prior work.

Also, here are several issues with the writing throughout the paper. For instance,
- "Even though, the function space of DNN is large," in the introduction
- "studying the generalization of DNNs by exploring property of SGD," in related work
- " the value of concerning" in section 3
- "restricttion" in section 3
- "popular gradient" in section 3


**Questions:**

Is there some way to at least estimate $\gamma'$ and $\mathbb{V}_m$?

Relatedly, how does the tightness of the bounds produced by this method compare to other practical generalization bounds?

**Limitations:**

Limitations are adequately addressed; the authors devote an entire section of the paper to limitations. No negative potential societal impacts.

---

> ### Author Rebuttal · Authors · 2023-08-06
>
> Thanks for your positive comment.
>
> In the overall responce, we device a toy dataset to compare our results with previous works. We will check and fix the typos in the paper.

---

### Official Review · Reviewer_t8yd · 2023-07-06

**Soundness:** 2 fair
**Presentation:** 3 good
**Contribution:** 2 fair
**Rating:** 5
**Confidence:** 3

**Summary:**

This paper studies the generalization of a general function class under the (S)GD algorithm with minimal assumptions.
Specifically, it gives a generalization (upper) bound based on several training trajectory characters during training: variance of the gradients, gradient norm,  training loss values and the learning rate.
Numerical experiments are conducted to verify the assumption and the generalization bound.

**Strengths:**

* The paper is written clearly and the ideas have been presented in the proper order.

* This paper studies the generalization for a very general class function that not only includes for neural network models, which can provide more insights in other non neural network based scenarios.

**Weaknesses:**

The major weakness is that this paper does not go into detail about studying the asymptotic order of the generalization bound in Theorem 3.6.
To show the superiority of this new proposed method in this paper, it is very important to provide (at least) the asymptotic analysis on the generalization bound with respect to the sample size $n$.

For example, in which case does the model benign overfitting (generalization gap goes to zero as $n \xrightarrow{} \infty$)?
Can this new generalization bound cover (or outperform) previous bounds in some specific settings, e.g., linear regression, kernel regression, or overparameterization neural networks?

In the current version of this paper, even though it provides a general framework for generalization analysis, I cannot see the potentiality of this generalization bound.

---

Another weakness is for the Assumption 3.4.
In fact, this assumption is trivially hold for finite set $\mathrm{w}$ because we can always take the supremum $\sup_{w \in \mathrm{w}} \|\nabla F_{\mu}(w)\|$ and the infimum $\inf_{w \in \mathrm{w}} \|\nabla F_{S}(w)\|$ and then get $\gamma$.
However, once the set $\mathrm{w}$ contains a point that is a stationary point of the empirical risk, i.e., $\|\nabla F_{S}(w^*)\| = 0$ for some $w^* \in \mathrm{w}$, then Assumption 3.4 implies that $w^*$ is also a stationary point of the population risk $F_{\mu}(w)$, which is a very restrictive property as the data distribution matters.
It is also shown in the Figure 1 that as the training loss goes to zero (epoch increases), the $\tilde{\gamma}_t$ may diverge.

That is to say, the relaxed Assumption A.7 is more reasonable.


**Questions:**

For Theorem 3.6, the optimal generalization bound one can get is $\mathcal{O}(\eta_m)$, where $\eta_m = \max_t \eta_t$.
This means that to get a non-vacuous bound, $\eta_m$ should decay to zero with respect to the sample size $n$, but the sample size dependent learning rate is not common in the practical setting, so why there is such a term?
Compared with the uniform-stability based generalization bound $\mathcal{O}\(\frac{\sum_{t=1}^T \eta_t}{n}\)$ in *Hardt eta al.*,  we can see that as long as $T = o(n)$, constant step-size SGD is provably to having a non-vacuous bound, so what is the gap here?

**Limitations:**

This paper proposes a framework on the generalization analysis for a wide class of function, but more works are still needed to be done to show the superiority of this new method.

As mentioned in the weakness part, I think this paper can benefit from the following two aspects:
* more asymptotic analysis on the generalization bound in common machine learning settings, such as linear regression, kernel regression and overparmeterized neural networks.
* since the bound in this paper does not require that $f$ is a neural network model, I think it would be more convincing if the authors could add some examples to show the generalization of other non neural network models, such as gradient boosting.

---

> ### Author Rebuttal · Authors · 2023-08-06
>
> Thank you for your time. The two points you raised (1. Analysis on machine learning settings and 2. Analysis on non-neural network models) are indeed interesting and beneficial for a deeper understanding of our proposed method.
>
> ## Assumption:
>
> The primary objective of our paper is to examine the generalization behavior of overparameterized neural networks trained using GD/SGD, which is a challenging issue. In the case of overparameterized neural networks, reaching a stationary point is a rare occurrence, as shown in reference [41] and validated in Figure 1. For general machine learning cases, this is a strong assumption. Consequently, we introduce the relaxed Assumption A.7 and **its corresponding bound (Theorem A.8)** in the Appendix.
>
> ## Question:
>
> 1. For the bound in Hardt eta al, the batch size used in their proof is 1. This means that the $T$ is usually large in setting. If we use the learning rate as fixed $\eta$ and training for $e$ epochs. Then $\mathcal{O}(\frac{\sum_{t=1}^T \eta_t}{n})=\mathcal{O}(\frac{e \times n \times  \eta}{n})=\mathcal{O}(e  \eta )$. Actually we have $\lim \limits_{n\to \infty} \mathcal{O}(\eta_m)=0$ (Proposition A.1 in The appendix.). However, we cannot figure out a concrete form like $\mathcal{O}(\frac{1}{n^c})$. This is a limitation of our current work.
>
>
> 2. The analysis of $n$. We will first analysis the dependent of $n$ for $\mathbb{V}$. The $\mathbb{V}$ is calculated as $\mathbb{V}(\mathbf{w})=\frac{\Vert \nabla F_S(\mathbf{w})\Vert}{\mathbb{E} _ {U \subset S} \Vert \frac{|U|}{n} \nabla F_U(\mathbf{w})-\frac{n-|U|}{n}\nabla F_{S/U}(\mathbf{w}) \Vert}$. Obvious, the gradient of individual sample is unrelated to the sample size $n$. And $\vert U \vert \sim n$. Therefore, $ \mathbb{V}=\mathcal{O}(1)$. Similarly, we have $\mathbb{E} \int_t \frac{d F_S(\mathbf{J_t})}{\sqrt{n}} \sqrt{1+\frac{\operatorname{Tr}(\Sigma(\mathbf{J_t}))}{\| \nabla F_S(\mathbf{J_t}) \|_2^2}}=\mathcal{O}(\frac{1}{\sqrt{n}}) $.  As for the
>  $\mathcal{O}(\eta_m)$ term in Theorem 3.6, we have $\lim \limits _ {n \to \infty} \mathcal{O}(\eta_m) =0$ according to Proposition A.1. We simply assume that $ \mathcal{O}(\eta_m)=\mathcal{O}(\frac{1}{n^c})$. Therefore, our bound has $\mathcal{O}(\frac{1}{n^{\text{min}\lbrace 0.5,c \rbrace}})$
>
> ## Two suggestions:
>
> #### 1 Linear Regression.
>
> We observe that our methods in linear regression tend to degenerate to a form resembling the traditional Rademacher complexity method. This is understandable, as our approach originates from Rademacher complexity and is more adept at analyzing complex neural networks.
>
> We denote the data as: $z_i \triangleq \lbrace x_i,y_i \rbrace$ and $S=\lbrace z_i \rbrace_{i=1}^n$,
> where the $x_i \in \mathbb{R}^{in}$ is the data and $y_i \in \mathbb{R}$ is the corresponding label.
> And the matrix of all data and labels are denoted as $\mathbf{x}\in \mathbb{R}^{in \times n}$, $\mathbf{y}\in \mathbb{R}^n$.
> The function $f(\cdot)$ is defined as $f(\mathbf{w},z_i)=\frac{1}{2}( y_i -\mathbf{w}^{\mathrm{T}} x_i ) ^2$.
> Therefore, we have $\nabla f(\mathbf{w},z_i)=(\mathbf{w}^{\mathrm{T}}x_i-y_i)x_i$. The weights after $t$-th update as $\mathbf{w} _ t$.
>
> The generalization error of our method has the form $\mathcal{O}(\int _ t  \frac{\Vert \mathrm{d} \mathbf{w} _ t \Vert}{\sqrt{n}}  \sqrt{F _ S(\mathbf{w} _ t) \text{max} _ i(\Vert x _ i\Vert)}$.
>
> For the Rademacher complexity, we have $\mathcal{O}(\frac{\Vert \mathbf{w}_T \Vert}{\sqrt{n}} \sqrt{\text{max}_t F_S(\mathbf{w}_t)\text{Tr}(\frac{1}{n}\mathbf{x}^{\mathrm{T}} \mathbf{x})})$.
>
> Therefore, on the linear regression, our bound is similar to that of Rademacher complexity.
>
> #### 2 Gradient boosting
>
> Gradient boosting is **beyond the scope of our paper**, as it focuses on function space while our method requires updates in weight space. Nonetheless, we offer a preliminary study using our framework.
>
> $F^{(j)}: \mathbb{R}^d \to \mathbb{R}$ is the ensemble of $j$ models. $l: \mathbb{R} \times \mathbb{R} \to \mathbb{R} _ {+}$ is the loss function. We consider a distant measure $d(\cdot,\cdot)$. We choose the function $f^{(j)}:\mathbb{R}^d \to \mathbb{R}$ from function space $\mathcal{F}^{(j)}$. We have $F^{(j)}=\sum _ {k=0}^{k=j-1} \delta _ k f^{(k)}$. We denotes $S=\lbrace(x _ i,y _ i)\rbrace _ {i=1}^n$ as our training data and  $S' = \lbrace( x' _ i,y' _ i)\rbrace _ {i=1}^n$ as the test data. We simplify the notation that $F^{(j)} _ i=F^{(j)}(x _ i)$, $F'^{(j)} _ i=F'^{(j)}(x' _ i)$, $f^{(j)} _ i=f^{(j)}(x _ i)$ and $f'^{(j)} _ i=f'^{(j)}(x' _ i)$. In the gradient boosting, we choose the $f^{(j)}$ such that $\sum _ i \frac{1}{n} d(f^{(j)} _ i,-\nabla l(F^{(j)} _ i,y _ i))$ is small. We define $\mathcal{L}=\sum _ {i,j} \frac{1}{n} \delta _ j d(f^{(j)} _ i,-\nabla l(F^{(j)} _ i,y _ i))$ and $\mathcal{L'}=\sum _ {i,j} \frac{1}{n} \delta _ j d(f'^{(j)} _ i,-\nabla l(F'^{(j)} _ i,y' _ i))$. Here we focus on analyzing the difference between $\mathcal{L}$ and $\mathcal{L}'$.
>
> We denote $g_i(f^{(j)})=g_i^{+1}(f^{(j)})=d(f^{(j)}_i,-\nabla l(F^{(j)}_i,y_i))$ and $g'_i(f^{(j)})=g_i^{-1}(f^{(j)})=d(f'^{(j)}_i,-\nabla l(F'^{(j)}_i,y'_i))$.
>
> We denfine two complexity measure of function space $\mathcal{F}$ based the distance measure: $R^a _ S(\mathcal{F})=\mathbb{E} _ {\sigma} \sup _ {f \in \mathcal{F}} \frac{1}{n} \sum _ {i=1}^n\sigma _ i g _ i $ and $R^b _ S(\mathcal{F})=\mathbb{E} _ {S,k}  \inf _ {f \in \mathcal{F}} \frac{1}{n} \sum _ {i,\sigma_i=-1} d(f _ i,f _ k)$.
> If samples of $S$ and $S'$ follow a same distribution, we have the following conclusion:
>
> $$\mathbb{E}[\mathcal{L'}-\mathcal{L}] < 2 \sum_j  \delta_j( R_S^a(\mathcal{F}^{(j)})+R_S^b(\mathcal{F}^{(j)})) + 2\sum_{j} \delta_j \mathbb{E} d(\nabla l(F_i^{(j)},y_i),\nabla l(F'^{(j)}_i,y'_i)) + \epsilon $$
>
> The first term is the complexity of the function space and the second term is to measure how the $F^{(j)}$ depends on the set $S$. The dependence exsits because the choose of $f^{(j)}$ is related to the $S$.

---

> > ### Comment · Reviewer_t8yd · 2023-08-12
> > **Rebuttal Update**
> >
> > I thank the authors for the rebuttal. I have decided to increase my score.

---

### Official Review · Reviewer_JRdZ · 2023-07-06

**Soundness:** 3 good
**Presentation:** 3 good
**Contribution:** 3 good
**Rating:** 6
**Confidence:** 3

**Summary:**

This paper present a new generation bound with moderate realistic assumptions that incorporate new information from gradients and trajectory of learning.

**Strengths:**

1 - the paper is well written and easy to follow.
2 - the paper does a great job comparing their results with previous literature on this topic.
3 - their experiments shows a promising result.

**Weaknesses:**

1- experiments are limited. This paper can definitely benefit from more experiments, however due to page limit, it is too much to ask.

**Questions:**

see above.

**Limitations:**

The paper address their limitation in their paper well.

---

> ### Author Rebuttal · Authors · 2023-08-06
>
> Thank you for taking the time to review the paper.
>
> We have included additional experiments in the overall response, featuring a toy dataset to compare tightness, as well as experiments on ResNet18 with CIFAR-10 and WikiText-2 datasets.

---

### Official Review · Reviewer_dakY · 2023-07-06

**Soundness:** 4 excellent
**Presentation:** 4 excellent
**Contribution:** 3 good
**Rating:** 6
**Confidence:** 3

**Summary:**

This work proposed a novel generalisation error bound which takes the learning trajectory of neural networks into consideration.
Instead of focusing on the post-trained neural networks, the proposed bound bases on the parameter updates during the learning of the neural networks.
The core proof of the bound relies on the assumption 3.4, as well as decomposing the difference between the updates on the true generative distribution and the training instances, i.e. $F\_{\mu}(\mathbf{J}\_{T}) - F\_{S}(\mathbf{J}\_{T})$, into a linear part $\mbox{gen}^{lin}(\mathbf{J}\_{T})$ and non-linear part $\mbox{gen}^{nl}(\mathbf{J}\_{T})$.
With reasonable assumptions, the authors bound the non-linear part with $\mathcal{O}(\eta\_{m})$.
Regarding the upper bound of the linear part, it relies on the other assumptions introduced in Section 3.2.


**Strengths:**

1. The main result, i.e. a generalisation bound that takes the learning trajectory information, is novel and interesting.
Practical exploration also demonstrates that the learning trajectory information helps neural networks to obtain better generalisation performance.
For example, knowledge distillation [1] can be considered a method to update the original labels to incorporate the predictions from the teacher model.
There also exists works that directly modify the labels with learning trajectory information, e.g. [2], and they have shown that the learning trajectory information are indeed helpful for generalisation.
So, a theoretical work that can bound the generalisation errors with learning trajectory information is interesting to the community.

2. The proof is well sketched, thus relatively straightforward to follow: not only the sketch from line 227 to line 238, also the formal proof in Appendix A.1.

3. The authors have also provided empirical evidence to show that the core assumption 3.4 hold, as well as how the generalisation error varies during the learning of neural networks.

4. The overall structure of the paper is clear, and the structure is also well-organised (excluding Section 1, my reasons are given below).

[1] Hinton, Geoffrey, Oriol Vinyals, and Jeff Dean. "Distilling the knowledge in a neural network." arXiv preprint arXiv:1503.02531 (2015).
[2] Ren, Yi, Shangmin Guo, and Danica J. Sutherland. "Better supervisory signals by observing learning paths." arXiv preprint arXiv:2203.02485 (2022).

**Weaknesses:**

### Major

1. **Displaying the main result at the beginning**:
This is not a concern about the technical side of this work, but rather a comprehensibility issue.
Without reading Section 3, the meanings of the notations in Equation 1 are mysterious, and I cannot interpret it.
From my perspective, the Section 1.1 does hinder the comprehensibility of this work.

2. **Lack of necessary details of the experiments in Section 4**:
After reading the supplementary materials, there are still some necessary details of the experiments in Section 4 I didn't find.
For example, in line 284, the authors specify $S'$ to be another data set.
However, since the data distribution of CIFAR is unknown, I can only assume that both $S$ and $S'$ are subsets of CIFAR, whereas I'm not certain about that.
To fully reproduce these experiments, more details are necessary.

3. **Experiment in line 271 on Gaussian data**:
The authors omit $\gamma'$ due to the lack of the true data distribution in line 271.
This can be relatively easily solved by sampling training instances from a Gaussian distribution.

### Minor

1. The first $w$ in line 148 is not bold.

2. The font size of all figures is too small.
(I understand that the authors may want to save some space, but the texts are really too small to read.)

**Questions:**

My questions are pretty much the same as the weakness I listed above.

The only suggestion I want to raise is to move Section 1.1 after Section 3.2.

**Limitations:**

Yes, the authors have explicitly and clearly pointed out the limitations of this work.

---

> ### Author Rebuttal · Authors · 2023-08-05
>
> Thank you for your positive feedback. Based on your suggestions, we propose the following improvements to the paper:
>
> Major 1: Indeed, Section 1.1.1 may hinder the paper's understandability. We plan to move Section 1.1.1 to a location after Theorem 3.6 and Remark 3.7.
>
> Major 2: We will provide additional details about the experiments. The $S$ and $S'$ represent the training set and test set of the dataset, respectively. We will include this information to enhance the paper's readability.
>
> Major 3: We appreciate your advice. Conducting experiments on a Gaussian dataset is an excellent suggestion. We designed the experiments and discovered that the $\gamma$ is linearly related to the generalization error, which aligns with our experimental findings. Additionally, we compared the tightness of our bound with previous stability-based bounds. The results are presented in the overall response.
>
> Minor 1: We will make the necessary revisions.
>
> Minor 2: Thank you for pointing this out. Indeed, the font is too small. We will modify all images in the figures to resemble the format used in Figure BC in the PDF of the overall response.

---

> > ### Comment · Reviewer_dakY · 2023-08-19
> >
> > I confirm that I've read through the rebuttals from the authors. The updates look good! Good luck!

---

### Official Review · Reviewer_tZu2 · 2023-07-26

**Soundness:** 3 good
**Presentation:** 3 good
**Contribution:** 3 good
**Rating:** 6
**Confidence:** 4

**Summary:**

This paper studies the connection between the learning trajectories of DNNs and their generalization when optimized using SGD. Its main contribution is that it provides a good perspective for generalization error analysis by studying the contribution of the learning trajectory. Based on their analysis of the learning trajectory, a new generalization bound is provided for DNNs.

**Strengths:**

On the whole, the theoretical analysis perspective and ideas of this paper have a certain value for the theoretical research work of deep neural networks. The main strengths are:
1. It provides a perspective from the learning trajectory for the theoretical analysis of deep neural networks.
2. The proposed generalization error bound track changes in adjustments of learning rate and noise level.


**Weaknesses:**

1.	It is unknown how tight the generalization error bounds given in the paper are. Furthermore, the impact of commonly used learning rate schedulers (e.g., exponential decay schedulers) on generalization error bounds has not been adequately analyzed.
2.	The description of the insight of the theorem is not detailed enough. For example, the insight that Theorem 3.6 can bring to the reader is not presented in detail. In addition, what insights can Corollary 3.8 bring to readers?
3.	The description of the experiment in the paper is not detailed enough. For example, the neural network models used for Cifar-100 and SVHN are not indicated.
4.	The experiments done were not extensive enough. I'm curious to know whether the proposed theorem is also applicable to other commonly used neural network structures like ResNet, not just VGG.

**Questions:**

1.	Can the authors give an analysis and description of how tight the proposed generalization error bound is? Under what conditions does the generalization error inequality in Theorem 3.6 take the equal sign? Besides, how does the learning rate scheduler affect generalization error?
2.	Equation (11) in Corollary 3.8 is confusing. In this formula, both the integral of the variable t and the sum of t are included, which makes it unclear to the reader that the author regards the learning trajectory as a continuous variable (maybe the reason is that the learning rate is assumed to be sufficiently small?), or treat the trajectory as a discrete variable?
3.	Can the author describe in detail the insights Theorem 3.6 and Corollary 3.8 bring to the reader? What insights can these theorems bring to researchers about the optimization of neural network algorithms?
4.	Can the authors describe their experiments more specifically? For example, a more specific description of the neural network model used for different datasets.
5.	Why didn't the author show the test accuracy of the trained neural network model on different datasets? Can the test accuracy be put in an appendix to show that the experimentally obtained accuracy is reasonable or acceptable?
6.	Is the proposed theorem (generalization error bound) also suitable for NLP tasks? The supporting material provided by the author contains the relevant code of WikiText2, but the relevant results do not seem to be shown in the paper. Can the authors show experimental results on the WikiText2 dataset?
7.	Authors need to correct the full text. In some sentences, there is a missing space between the content of the paper and the index of the reference, such as "experiments[9, 13] have" in line 40 of the paper. Besides, Equation 38 is incomplete.
8.	Why \Delta FS (Jt)- \Delta FS(Jt) (line 293 of the paper) can be used to approximate the generalization error? It is different from the generalization error in formula (10). What is the author's reason for using it instead of generalization error in Equation (10) in the experiments?
9.	The symbols of L2 norm in the paper are not unified. Some formulas do have punctuation.


**Limitations:**

The authors describe their limitation as their method requiring small learning rates. How to eliminate this assumption is a direction worth studying in the future.

---

> ### Author Rebuttal · Authors · 2023-08-05
>
> Question 1 (Weakness 1): We provide a comparison on a toy dataset due to the challenges in calculating the bound, and the results are displayed in the overall response. Although we cannot currently determine the exact conditions for the equal sign, we will continue to investigate. Identifying such conditions is a difficult problem, even for widely accepted stability-based generalization bounds. In Appendix A.6, we analyze the influence of the learning rate on the generalization bound. A larger learning rate benefits generalization by pushing $\mathbf{J_t}$ into a position where $\operatorname{Tr}(\Sigma(\mathbf{J_t}))$ is small. In this regard, exponential decay schedulers may improve generalization compared to using a small learning rate throughout the training process. An initially larger learning rate can result in a relatively small $\operatorname{Tr}(\Sigma(\mathbf{J_t}))$. In the later phase, even if $\operatorname{Tr}(\Sigma(\mathbf{J_t}))$ becomes large due to the small learning rate, the term $d F_S(\mathbf{J_t})$ is small at this stage. As a result, it will reduce the term $- 2 \gamma' \mathbb{V}_m \mathbb{E}\int _t \frac{d F_S(\mathbf{J_t})}{\sqrt{n}} \sqrt{1+\frac{\operatorname{Tr}(\Sigma(\mathbf{J_t}))}{\Vert \nabla F_S(\mathbf{J_t}) \Vert _ 2^2}}$ in our bound.
>
>
>
> Question 2: The trajectory is considered as a discrete variable due to the nature of the update method (SGD). The integral is merely a simplification of the symbol. We denote this in line 136. Indeed, it is confusing to use these two types of symbols. To unify them in the paper, we will rewrite
> $\gamma' \mathbb{V}_m M_4^2 \sqrt{ \mathbb{E} \sum_t \frac{1}{n \beta^2  (t+1)^4} \left(1+\frac{\operatorname{Tr}(\Sigma(\mathbf{J_t}))}{\| \nabla F_S(\mathbf{J_t}) \|_2^2}\right)} $ in collary 3.8 as $\gamma' \mathbb{V}_m M_4^2 \sqrt{ \mathbb{E} \int \frac{1}{n \beta^2  (t+1)^4} \left(1+\frac{\operatorname{Tr}(\Sigma(\mathbf{J_t}))}{\| \nabla F_S(\mathbf{J_t}) \|_2^2}\right) \mathrm{d}t}$.
>
> Question 3 (Weakness 2): The primary insights of Theorem 3.6 are described in Section 1.1.1. Our generalization bound reveals the relationship between the "Bias of Training Set," "Diversity of Training Set," "Complexity of Learning Trajectory," and generalization error. The "complexity of learning trajectory" is associated with the gradient norm, gradient covariance, and training loss during the learning trajectory. The placement of Section 1.1.1 is not ideal, so we plan to move its content to a position closer to Theorem 3.6 to make it more accessible to readers. Regarding Corollary 3.8, we aim to provide a comparison between our proposed method and the stability-based method by presenting the bound with the learning rate schedule used by the stability-based method. We will add the comparison result from question 1 in the overall response near this corollary.
>
> Question 4: The architecture used for CIFAR-10, CIFAR-100, and SVHN is VGG13. We will include these descriptions in Appendix A.5.
>
> Question 5 (Weakness 3): The test accuracy for CIFAR-10, CIFAR-100, and SVHN are 87.64%, 55.08%, and 92.80%, respectively. Thank you for pointing this out. We will add these details to the Appendix.
>
> Question 6 (Weakness 4): The results on WikiText-2 are provided in Figure C of the PDF of the overall response. Additionally, we have conducted experiments with ResNet-18 on CIFAR-10, which can be found in Figure B of the overall response. We will include these results in the Appendix. The training config of ResNet18 on Cifar10 is the same as the experiment of VGG. For the config on WikiText2, the Transformer is 2 layers, 2 head and 200 embedding size. The model trained using SGD with learning rate 0.001, batch size 20
>
> Question 7: Thanks for point out. I will fix it in the new version of paper.
>
> Question 8: Apologies for the unclear statement in the paper. Our focus is on analyzing the value of $F_{\mu}(\mathbf{J_T}) - F_S(\mathbf{J_T})$. However, we cannot calculate $F_{\mu}(\mathbf{J_T})$ due to the unknown distribution $\mu$. One approach to address this issue is to design an unbiased estimate in the experiment. Since $\mathbf{J_t}$ is dependent on $S$, we can sample a new dataset $S'$, making $\mathbf{J_t}$ independent of $S'$. Based on this, we have $\mathbb{E}F_{S'}(\mathbf{J_T}) = F_{\mu}(\mathbf{J_T})$, which implies that $F_{S'}(\mathbf{J_T})$ is an unbiased estimate of $F_{\mu}(\mathbf{J_T})$. This aligns with the practical wisdom of using the gap between the performance on the test set and the performance on the training set as a measure for generalization behavior.
>
> Question 9: Thank you for bringing this to our attention. Indeed, there are some issues with the symbols in the paper. We will address and fix these errors.

---

> > ### Comment · Reviewer_tZu2 · 2023-08-16
> >
> > I have read through the author's response, and my concerns are addressed appropriately.

---

### Author Rebuttal · Authors · 2023-08-05

We thank ACs, SACs, PCs, and reviewers for the efforts and time spent in handling our paper.

Figures in the pdf. A: Result of toy dataset for Question 2 below.  B: Results of ResNet18 on Cifar10. C: Results of Transformer on WikiText2.

The training config of ResNet18 on Cifar10 is the same as the experiment of VGG. For the config on WikiText2,  the Transformer is 2 layers, 2 head and 200 embedding size. The model trained using SGD with learning rate 0.001, batch size 20.


Question 1: Tightness of the proposed bound

In a toy dataset setting, we compare our generalization bound with stability-based methods. We choose a toy dataset because calculating $\beta$ (under the $\beta$-smooth assumption) and $L$ (under the $L$-Lipschitz assumption) in stability-based work, as well as the values of $\mathcal{V}$ and $\gamma$ in our proposed bound, is challenging. Additionally, stability-based methods require a batch size of 1. In the following, we discuss the construction of the toy dataset used to compare the tightness of the generalization bounds.

 The training data is $X_{tr}=\lbrace x_i \rbrace_{i=1}^n$. All the data $x_i$ is sampled from Guassian distribution $\mathcal{N}(0,\mathbf{I}_d)$.

Sampling $\tilde{\mathbf{w}} \sim \mathcal{N}(0,\mathbf{I}_d)$,the ground truth is generated by $y_i=1 \ \ \text{if} \ \ \tilde{\mathbf{w}}^{\mathrm{T}} x_i>0 \ \ \text{else} \ \ 0$.

 The weights for learning is denoted as $\mathbf{w}$. The predict $\tilde{y}$ is calculated as $\tilde{y}_i =\mathbf{w}^{\mathrm{T}}x_i $.

 The loss for a simple data point is $l_i=\left\Vert y_i- \mathbf{w}^{\mathrm{T}}x_i \right\Vert_2$. The training loss is $\mathcal{L}=\sum_{i=1}^n l_i$. The test data is $X_{te}=\lbrace x'_i \rbrace$, where $x'_i= \tilde{x}'_i$ and $\tilde{x}'_i \sim \mathcal{N}(0,\mathbf{I}_d)$.

We evaluate the tightness of our bound by comparing our results with those in references [11] and [42] from the original paper. We set the learning rate as $\eta_t=\frac{1}{\beta t}$. Our reasons for comparing with these two papers are: 1. [11] is a representative study, 2. Both papers have theorems using a learning rate setting of $\eta_t=\mathcal{O}(\frac{1}{t})$, which aligns with Corollary 3.8 in our paper, and 3. They do not assume convexity. We use 100 samples for training and 1,000 samples for evaluation. The model is trained using SGD for 200 epochs. The generalization bounds we compare include Corollary 3.8 from our paper, Theorem 3.12 from [11], and Theorem 5 from [42].

Our results are:

Gen Error | Ours | [11] | [42]
|-------|-----|------|------|
1.49 | 3.62| 4.04 | 4417.00



Our bound is tighter under this setting. The reason for the value of [42] is large is because that our and [11] has dependent on $\frac{L^2}{\beta}$, while [42] depends on $L^2$. $L$ and $\beta$ are usually large numbers. (Note that in corollary 3.8 of our works, we can replace $M_2$ and $M_4$ with $L$ because $M_2\leq L$ and $M_4 \leq L$ and $c$ in Theorem 3.12 of [11] is equal to $\frac{1}{\beta}$ with our setting.).

Question 2. Effect of $\gamma$

Under the aforementioned settings, we modify the variance of test data samples such that $\tilde{x}'_i \sim \mathcal{N}(0,\kappa \mathbf{I}_d)$. By using different values of $\kappa$, we obtain various values of $\gamma$ and discover a linear correlation between $\gamma$ and generalization error (Figure A in the rebuttal PDF). This finding is consistent with our theorem.

[11] M. Hardt, B. Recht, and Y. Singer. Train faster, generalize better: Stability of stochastic gradient 352 descent. In International conference on machine learning, pages 1225–1234. PMLR, 2016.

[41] J. Zhang, H. Li, S. Sra, and A. Jadbabaie. Neural network weights do not converge to stationary points: An invariant measure perspective. In International Conference on Machine Learning, pages 26330–26346. PMLR, 2022.

[42] Y. Zhang, W. Zhang, S. Bald, V. Pingali, C. Chen, and M. Goswami. Stability of sgd: Tightness analysis and improved bounds. In Uncertainty in Artificial Intelligence, pages 2364–2373. PMLR, 2022

---

### Decision · Program_Chairs · 2023-09-21

**Decision:**

Accept (poster)

**Comment:**

This paper studies a novel generalization bound based on the complexity of the function space that can be explored during the training process. The primary motivation is to consider the trajectory information during training and use that to get tighter generalization results. The reviewers questioned whether the bound improves upon classical stability-based generalization results. The authors used a toy example to demonstrate the improvement. Reviewers also asked whether the bound can show the convergence of the generalization error when the number of samples goes to infinity. The author’s rebuttal also addresses this by analyzing the asymptotic rate of the generalization error.